# Yu-Shiba-Rusinov bands in a self-assembled kagome lattice of magnetic molecules

Laëtitia Farinacci [1] ✉, Gaël Reecht[1], Felix von Oppen [2] & Katharina J. Franke [1]

Kagome lattices constitute versatile platforms for studying paradigmatic correlated phases. While molecular self-assembly of kagome structures on metallic substrates is promising, it is challenging to realize pristine kagome properties because of hybridization with the bulk degrees of freedom and modified electron-electron interactions. We suggest that a superconducting substrate offers an compelling platform for realizing a magnetic kagome lattice. Exchange coupling induces kagome-derived bands at the interface, which are protected from the bulk by the superconducting energy gap. We realize a magnetic kagome lattice on a superconductor by depositing Fe-porphin-chloride molecules on Pb(111) and using temperature-activated de-chlorination and self-assembly. This allows us to control the formation of smaller kagome precursors and long-range ordered kagome islands. Using scanning tunneling microscopy and spectroscopy at 1.6 K, we identify Yu-Shiba-Rusinov states inside the superconducting energy gap and track their hybridization from the precursors to larger islands, where the kagome lattice induces extended YSR bands. These YSR-derived kagome bands inside the superconducting energy gap allow for long-range coupling and induced pairing correlations, motivating further studies to resolve possible spin-liquid or Kondo-lattice-type behavior.

Kagome lattices are made of hexagonal tiles and corner-sharing triangles. Atomic lattices of this type feature many remarkable physical properties[1]. Their band structure is characterized by strongly dispersive states with Dirac cones originating from bonding and anti-bonding states in the hexagonal lattice co-existing with a flat band resulting from the self-localization of an additional electronic state. Flat bands are of great general interest, as they are prone to strong many-body correlation effects[2,3], whereas the Dirac cones enable the study of relativistic effects[4]. The richness of physical phenomena increases further when spin degrees of freedom with anti-ferromagnetic nearest-neighbor exchange interactions enter the stage. The crystal symmetry frustrates the spin interactions, favoring spin-liquid behavior[5,6], fractional excitations[5], an anomalous Hall effect[7,8], and chiral magnetic order[9]. Spin-orbit or many-body interactions may open a topological gap, expanding the class of topologically non-trivial materials[10].

So far, the study of kagome materials relied mostly on three-dimensional realizations, where stacks of kagome lattices form a bulk crystal. Inter-layer coupling alters the band structure and suppresses the effects of topology and correlations depending on the coupling strength. The investigation of pristine kagome properties requires monolayers with a kagome structure. A versatile approach to growing strictly two-dimensional kagome lattices makes use of molecular self-assembly on surfaces[11–14]. While the first kagome lattices consisted exclusively of closed-shell molecules, inserting magnetic atoms or molecules into the lattice introduces spin in addition to the electronic degrees of freedom.

However, despite the truly two-dimensional nature of the kagome lattice itself, its properties are strongly influenced by interactions with a bulk metallic substrate. Most notably, the substrate affects the electron–electron interactions[15]. We suggest that self-assembled kagome lattices are particularly promising when a superconducting

[1]Fachbereich Physik, Freie Universität Berlin, Arnimallee 14, 14195 Berlin, Germany. [2]Dahlem Center for Complex Quantum Systems and Fachbereich Physik, Freie Universität Berlin, 14195 Berlin, Germany. ✉e-mail: laetitia.farinacci@polytechnique.org

substrate is employed. Exchange-coupled spins on superconductors induce Yu–Shiba–Rusinov (YSR) states in the superconducting gap[16–18]. States from neighboring lattice sites may hybridize[18–23] and form extended bands inside the gap[24–26]. Hence, YSR states may be used to realize a kagome band structure, which has a truly two-dimensional character and is protected by the superconducting gap. The slow decay of the YSR states may induce long-range couplings and superconducting correlations may modify the normal-state kagome band structure. Such a system would thus provide a platform to study correlated phases of isolated kagome lattices. An even richer phase diagram emerges when the molecular spins couple (anti-ferro-) magnetically, leading to correlations between spin and electronic degrees of freedom[27] without loss of the two-dimensional character.

A recent experiment succeeded in growing a kagome lattice of Ni atoms on Pb(111), but charge transfer quenched the magnetic moments of the Ni atoms[28]. Yan et al.[29] reported self-assembly of a metal-organic kagome structure on superconducting NbSe$_2$ with kagome band formation at larger energies, but not within the superconducting energy gap. In contrast, other molecular assemblies on superconductors yielded YSR states inside the superconducting energy gap[30–34] but did not feature kagome structures.

Here, we show that under appropriate conditions, iron–porphin–chloride (FeP–Cl) (Fig. 1a) molecules self-assemble into a kagome lattice on a superconducting Pb(111) surface (Fig. 1b). We find that the molecule-induced YSR states (Fig. 1c) are hybridized within smaller kagome precursors and form extended bands in larger islands. We observe edge states at kagome domain boundaries but no signatures of topologically non-trivial states. In the normal state of Pb, the exchange coupling of the molecular spins to the substrate is reflected in a Kondo resonance. We suggest that kagome lattices on superconductors constitute a versatile platform for studying correlation effects owing to the protected band structure in the superconducting state while substrate-mediated magnetic interactions can be tuned, e.g., by varying the unit-cell size of the kagome lattice.

## Results and discussion
### Self-assembled molecular structures
We start by describing the self-assembly of FeP–Cl molecules on Pb(111). We find that we can tune the ratio $r$ between Cl adatoms and FeP molecules such that patches of different-sized kagome precursors (Fig. 2e,f) or a long-range ordered kagome lattice are formed (Fig. 2c, d). As we will show later, the analysis of YSR states in the various structures allows us to track the evolution of kagome bands.

In Fig. 2a, we show the molecular phase obtained after FeP–Cl deposition on a Pb(111) sample held at ≈230 K. The individual molecules appear either as clover shapes with a bright protrusion at their center or as almost rectangular shapes (close-up view in Fig. 2b). Some small protrusions are also observed between molecules (see arrow in Fig. 2b). We surmise that these protrusions are Cl atoms that have detached from FeP–Cl molecules and that both molecular types (clover shape and rectangle) correspond to de-chlorinated FeP molecules with different electronic configurations (see Supplementary Note 1). The FeP molecules and Cl atoms self-assemble into a hexagonal lattice

with the individual molecules being randomly oriented with respect to each other.

After annealing the low-temperature molecular phase to ≈370 K, all molecules display the clover shape, and we observe the formation of a kagome lattice (Fig. 2c). The lattice consists of hexagonal and triangular tiles, densely and periodically filling the surface. The triangular tiles are all occupied by a Cl atom, while the hexagonal tiles are either empty (blue arrow) or occupied by a FeP molecule (red arrow). On some parts of the same preparation, we observe imperfections of the kagome lattice (Fig. 2e). The periodicity of the lattice is broken into domains that still exhibit the structural motifs of triangular and hexagonal tiles (see red and black dashed lines). We will refer to these structures as kagome precursors. Not all of the triangular tiles are now occupied by Cl atoms (see red and blue triangles in Fig. 2f).

The unit cell of the defect-free kagome lattice, indicated in orange in Fig. 2c, consists of three FeP molecules and two Cl atoms within the triangular tiles. The ideal ratio of Cl atoms to FeP molecules is thus $r = 2/3$. If $r < 2/3$, the excess molecules are accommodated in the hexagonal tiles. However, once these tiles are filled at $r = 1/2$, the deficit of Cl atoms can no longer be compensated, and the long-range kagome lattice breaks up into smaller domains–the kagome precursors.

Deposition of the FeP–Cl molecules on a sample held above ≈300 K immediately leads to all molecules being de-chlorinated as evidenced by their clover-shape (Fig. 2g). The desorption barrier for Cl atoms can thus be overcome already during the adsorption process. The de-chlorinated molecules then assemble into large islands in which they arrange again in a hexagonal pattern with two molecular orientations, corresponding to a 45° rotation with respect to each other.

Concluding the structural analysis, the formation of the different molecular phases can be understood as a consequence of Cl desorption during deposition and annealing. Upon adsorption of FeP–Cl molecules on the Pb(111) substrate, the Cl ligands of the FeP molecules detach from the Fe centers. While they immediately desorb at high deposition temperatures, they remain on the sample at low temperatures and are mainly captured between the molecules. The degree of Cl capture on the surface can be fine-tuned by annealing. The Cl atoms are crucial for the formation of the kagome lattice. In particular, they seem to stabilize the triangular tiles. A high Cl-atom to FeP ratio $r$ ($r > 2/3$) prevents the formation of the kagome lattice as there would be too many triangular tiles (eventually leading to the hexagonal lattice at $r = 1$). After annealing, desorption of Cl atoms leads to a decrease of $r$, and a kagome lattice is observed for $2/3 \geq r \geq 1/2$. When $r < 1/2$, the excess FeP molecules can no longer be accommodated in the hexagonal tiles of the kagome lattice, and the long-range order of the kagome lattice is broken.

### YSR states of kagome precursors
Next, we characterize the magnetic properties of the FeP molecules in the kagome lattice and the smaller domains of only a few kagome tiles, i.e., of the kagome precursors. In particular, we are interested in the YSR states and their evolution from smaller to larger precursors. To probe the YSR states with high energy resolution, we use

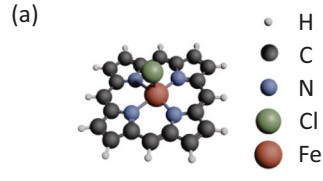
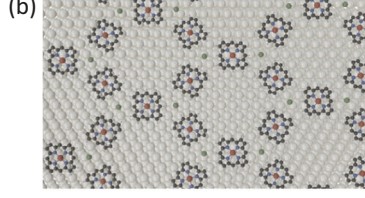
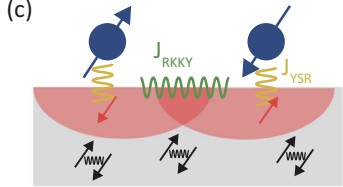

**Fig. 1 | Kagome lattice on a superconductor. a** Structure of a FeP–Cl molecule. **b** We use molecular self-assembly of FeP–Cl molecules on Pb(111) to obtain a 2D kagome lattice. **c** At each site, exchange coupling $J_{YSR}$ between the magnetic impurity and the substrate leads to a YSR state. Neighboring YSR states hybridize with one another. The impurity spins may also interact via RKKY interactions.

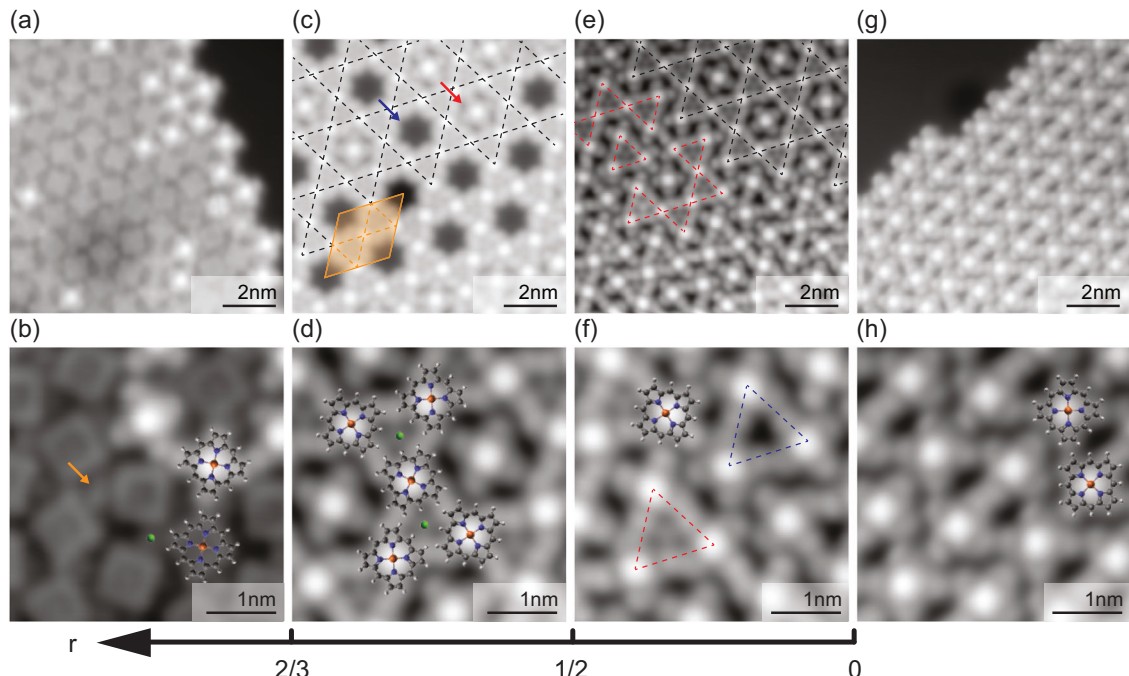

**Fig. 2 | Kagome structure formation. a** and **b** After deposition of FeP–Cl molecules on a Pb(111) sample held at ≈230 K, Cl is detached from the molecules but captured next to the FeP molecules ($V_{bias}$ = 20 mV, $I$ = 500 pA). After annealing to ≈370 K, the ratio $r$ of Cl adatoms to FeP molecules decreases, leading to the formation of (**c** and **d**) a kagome lattice for $1/2 < r < 2/3$ ($V_{bias}$ = 5 mV, $I$ = 50 pA) and of (**e** and **f**) kagome precursors for $r < 1/2$ ($V_{bias}$ = 5 mV, $I$ = 200 pA). **g** and **h** Deposition above ≈300 K leads to a hexagonal arrangement of the FeP molecules, i.e., no Cl remains on the surface ($V_{bias}$ = 5 mV, $I$ = 200 pA). Darker areas as in (**a**) indicate Ne inclusions below the surface.

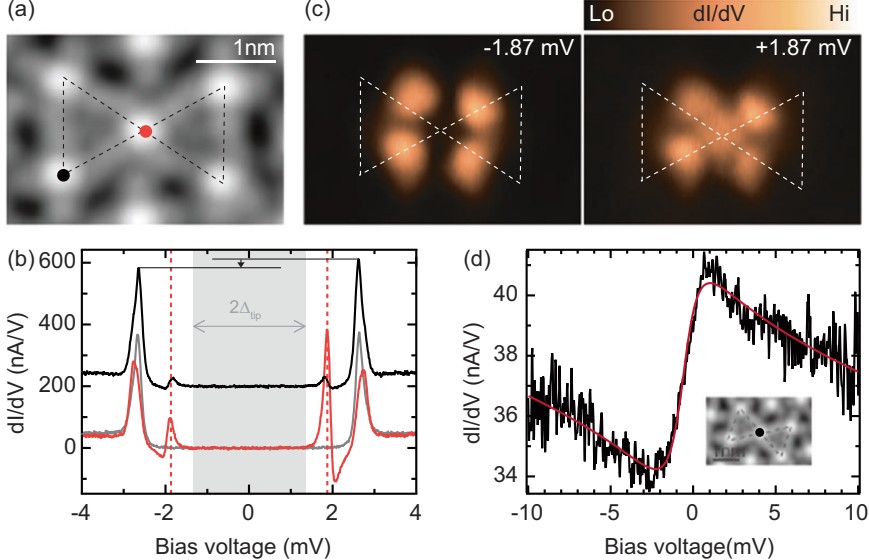

**Fig. 3 | YSR states and Kondo resonance of the smallest kagome precursor.**
**a** STM topography image ($V_{bias}$ = 5 mV, $I$ = 200 pA). **b** d$I$/d$V$ spectra recorded above a molecule at the edge of the structure (black, offset for clarity) and on the central molecule (red), along with a reference spectrum taken on the bare Pb surface (gray). Both spectra are recorded on the centers of the two molecules. **c** d$I$/d$V$ maps recorded at $V_{bias}$ = ±1.87 mV while the tip follows the height profile of the STM topography image in (**a**) ($V_{rms}$ = 25 μeV). The dashed lines are replicas of those in (**a**) to help identify the molecules' positions. The d$I$/d$V$ signal is present only above the central molecule. **d** Spectrum taken above the center of a similar kagome precursor and in the normal state of Pb by applying an external magnetic field $B_{ext}$ = 600 mT ($V_{rms}$ = 50 μeV). The red line is a fit to a Fano-Hurwitz function[40], yielding a Kondo temperature of $T_K$ = 7.9 ± 0.3 K.

superconducting Pb tips, which have been prepared by deliberately crashing the tip in the clean Pb surface. The bulk-like superconducting properties can be checked by a doubling of the superconducting energy gap in differential conductance (d$I$/d$V$) spectra.

According to the analysis of the structure in Fig. 2d, the two corner-sharing triangles, including Cl atoms, correspond to two triangular tiles of the kagome lattice. Such a kagome precursor (black dashed lines) is shown in Fig. 3a. The molecule at the center of the structure (red dot) is flanked by two Cl atoms (corresponding to the occupied tiles), while the molecules at the edge of the structure only have one adatom in their vicinity. These different surroundings translate to different signatures in their d$I$/d$V$ spectra (Fig. 3b). The

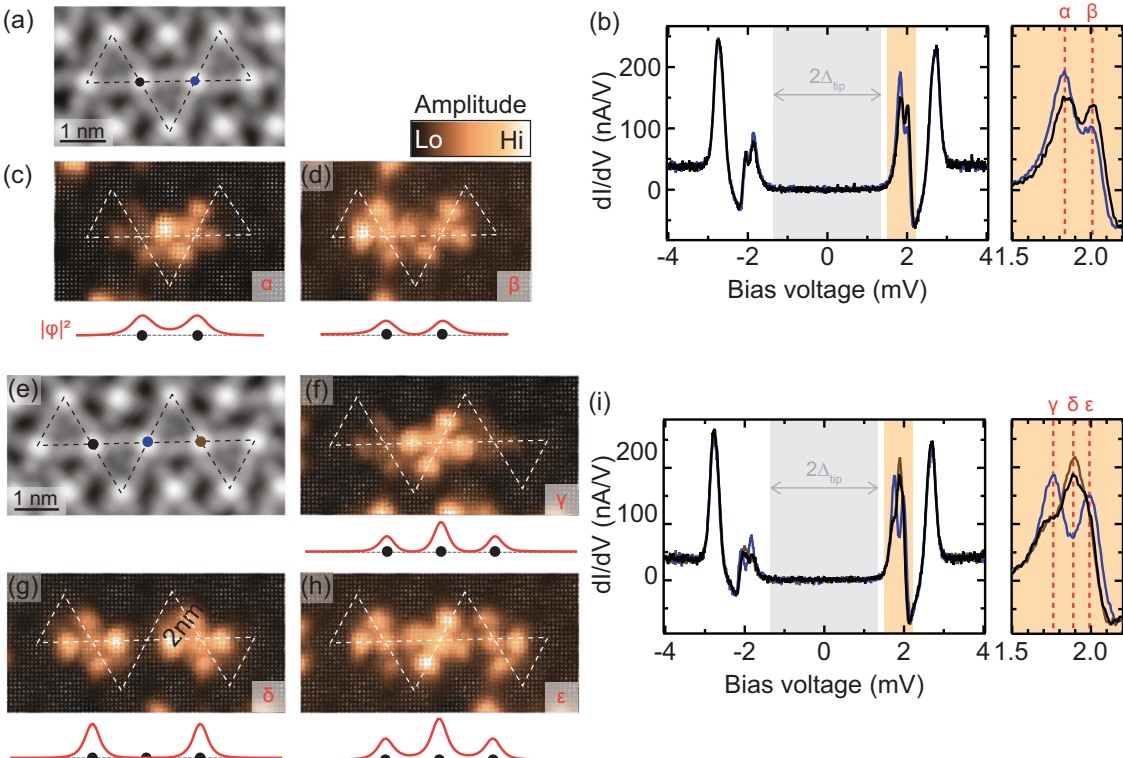

**Fig. 4 | Hybridization of YSR states in kagome precursors. a** Topographic image ($V_{bias} = 5$ mV, $I = 200$ pA) of a kagome precursor with two molecules that have two Cl adatoms in their vicinity. **b** d$I$/d$V$ spectra recorded above the centers of these molecules show the presence of two YSR states. The intensity of these states across the structure is obtained by recording a grid of d$I$/d$V$ spectra ($V_{rms} = 15$ μeV) as shown in **c** and **d** and fitting the amplitude of each state after deconvolution (see Supplementary Note 3). As sketched below each data panel, the intensity distribution of the states matches that of a two-site chain with nearest neighbor coupling. **e** Topographic image ($V_{bias} = 5$ mV, $I = 200$ pA) of a kagome precursor with three molecules that have two Cl adatoms in their vicinity. Three YSR states are now observed in d$I$/d$V$ (**i**) ($V_{rms} = 15$ μeV), and the amplitude distributions of the states shown in **f**–**h** match those of a three-site chain with nearest-neighbor coupling.

central molecule exhibits a pair of resonances at ±1.87 mV inside the superconducting energy gap of ±2$\Delta$ = ±2.7 mV, signaling the presence of a YSR state well inside the superconducting gap. Tip-approach measurements above the Fe center further reveal that the system is in a screened spin ground state[31] (see Supplementary Note 4). This is in accordance with measurements taken in an external magnetic field, which quenches superconductivity in the Pb substrate and reveals the presence of a Kondo resonance with a Kondo temperature of $T_K = 7.9 \pm 0.3$ K (Fig. 3d). This Kondo temperature indicates a coupling strength that is beyond the quantum phase transition to the screened spin regime that is expected to take place at $k_B T_K \sim 0.3\Delta$[35] with $\Delta = 1.36$ meV. The presence of a Kondo resonance also highlights the quantum nature of the spin. The molecules at the edge of the structure display an asymmetry of the coherence peaks, which we attribute to a YSR state whose energy is close to the pairing energy of the substrate. Molecules that are isolated on the surface (see Supplementary Note 7) or lie within a pore of the kagome lattice (see YSR band formation in the kagome lattice) also show only a weak asymmetry of the BCS coherence peaks. These types of spectra indicate that a single Cl atom does not strongly influence the position of the YSR states, in contrast to molecules with two Cl neighbors. We note that the faint resonances at ±1.87 mV originate from the extended YSR state of the central molecule, as one can see in the corresponding d$I$/d$V$ maps of Fig. 3c. The shape and asymmetry of the YSR state in d$I$/d$V$ measurements are related to interfering tunneling paths through the magnetic and frontier orbitals of the FeP molecule[36]. Essentially, the shape seen in the d$I$/d$V$ maps thus resembles the molecular orbitals rather than the wave function of the YSR states in the substrate. The Cl atoms do not induce YSR states. Instead, they act as a local gate modifying the coupling

strength similar to electrostatic fields imposed by neighboring molecules[32] or modulations in electronic potential due to a charge-density wave[37].

Importantly, the YSR states on the center molecule are sharp and do not display indications of hybridization with other YSR states despite the close vicinity of the corner molecules. We attribute the absence of hybridization to the large energy difference of the YSR states induced by the different number of neighboring Cl atoms. Hence, the presence of the Cl atoms is not only important for stabilizing the kagome structure but also for tuning the energy alignment of the YSR states. We note that we do not find indications of hybridization of YSR states of molecules at the edge of the structure (see Supplementary Note 2). However, even if there was weak hybridization, it would occur at energies well away from those that pertain to the molecules within the lattice, which we describe in the following.

## Hybridization of YSR states

Hybridization of YSR states can be observed in larger structures of triangular tiles, where each triangle shares a vertex with another one. This implies that several neighboring FeP molecules have two Cl adatoms in their vicinity and thus exhibit identical YSR states. Figure 4a and e show topographic images of kagome precursors (black dashed lines) with two and three such molecules, respectively. The d$I$/d$V$ spectra recorded above their Fe centers (as indicated by dots in the topographic images) show a splitting of the YSR state into two (Fig. 4b) or three (Fig. 4i) resonances, with intensities depending on the probed molecule. The splitting is a strong indication of YSR hybridization[20,21,25,38]. To confirm the coupling of YSR states within these structures, we unravel the spatial distribution of the YSR states

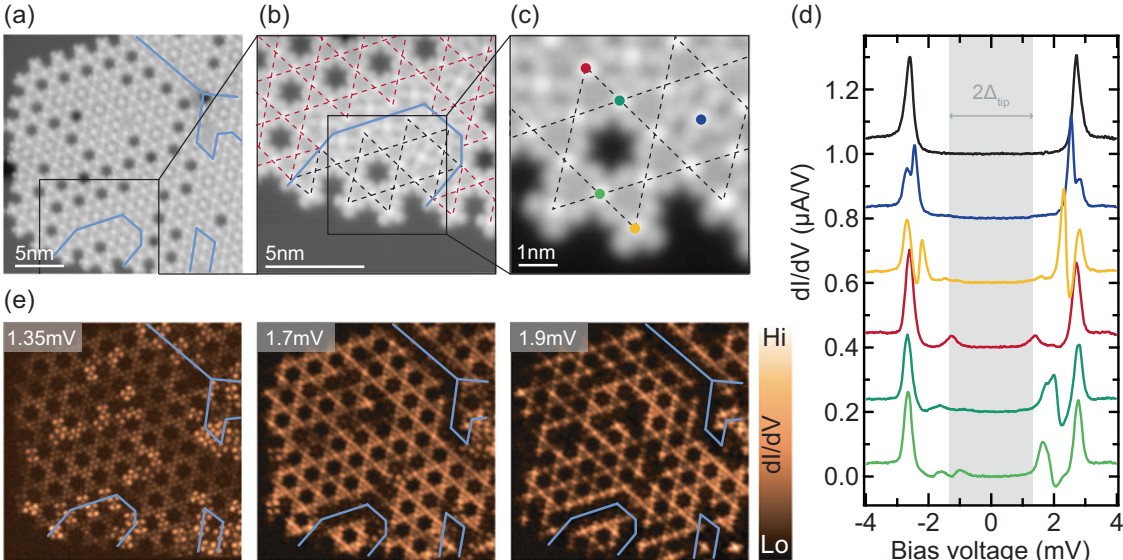

**Fig. 5 | Kagome bands and kagome boundaries. a** Topographic image ($V_{bias}$ = 5 mV, $I$ = 50 pA) of a molecular island in which several kagome domains coexist, with domain boundaries indicated by blue lines. **b** Enlarged view of such a domain ($V_{bias}$ = −45 mV, $I$ = 100 pA). **c** Molecules that have different positions relative to the kagome domains ($V_{bias}$ = −45 mV, $I$ = 100 pA) display **d** different fingerprints in d$I$/d$V$ spectroscopy ($V_{rms}$ = 20 µeV). A reference spectrum taken on bare Pb is shown in black. Spectra are offset for clarity. **e** Constant-height d$I$/d$V$ maps taken at bias voltages indicated in the top left corner ($V_{rms}$ = 25 µeV).

by recording d$I$/d$V$ spectra along densely spaced grids. We remove the contribution of the tip density of state (DOS) by numerical deconvolution of the d$I$/d$V$ spectra. We subsequently obtain the amplitude of each YSR state by fitting the sample DOS by a sum of Lorentzians (see Supplementary Note 3). The extracted amplitudes of the YSR states are then plotted in Fig. 4c, d and f–h (the dashed lines serve as guide-to-the-eye by indicating the structures of the kagome precursors determined from the simultaneously recorded topographic images).

The spatial intensity distribution along the molecular structure can be qualitatively reproduced by a simple tight-binding model with nearest-neighbor coupling between sites. A sketch of the expected intensity distribution is shown below each data panel. In analyzing the intensity distribution, we need to keep in mind that the shape seen in the d$I$/d$V$ maps is dominated by co-tunneling through molecular orbitals. Yet, it is possible to identify increased intensities and nodal planes between the kagome building blocks. More precisely, the kagome precursor of Fig. 4a shows a YSR state (labeled $\alpha$) with the largest intensity between the vertices of the triangle, i.e., between two FeP molecules (Fig. 4c), whereas the YSR state labeled $\beta$ has its intensity maxima at the vertices and a nodal plane in between (Fig. 4d). These YSR states thus match anti-symmetric and symmetric linear combinations of the YSR wavefunctions of the individual units (compare to Fig. 3c). Correspondingly, the kagome precursor of Fig. 4e exhibits hybridized YSR states that concord with those of a three-site chain. One state is mainly localized above the central site (Fig. 4f, YSR state $\gamma$), another one above the ends of the chain (Fig. 4g, YSR state $\delta$), and the last one is distributed over all three sites (Fig. 4h, YSR state $\epsilon$). These results evidence the hybridization of YSR states when the smallest precursors of two corner-sharing triangles are assembled into larger structures.

To understand whether the spins associated with the YSR states can give rise to interesting magnetic properties in the extended kagome lattice, we need to search for magnetic coupling between the units. By analyzing the shift of the YSR state upon approach with the STM tip, we conclude that the ground state is a screened-spin state (see Supplementary Note 4). A fully screened spin would not be available for magnetic coupling. However, in the gas phase, the FeP molecule carries a spin of $S$ = 1. If screening occurs only in one channel,

the ground state remains a spin-1/2 system, which could couple via Ruderman–Kittel–Kasuya–Yosida (RKKY) interactions. In contrast to our observations, strong RKKY coupling would lead to deviations from simple tight-binding chain behavior[27]. This suggests that in the present system, RKKY coupling is small compared to the hybridization energy.

### YSR band formation in the kagome lattice

Now that hybridization is established in kagome precursors, we examine the formation of YSR bands in a kagome lattice. The molecular island in Fig. 5a exhibits several kagome domains delineated by blue lines. An enlarged view of such a domain boundary is shown in Fig. 5b, where the two kagome lattices, indicated by red and black dashed lines, are mismatched by one molecular row. A close-up view of the black-dashed kagome domain is shown in Fig. 5c. Molecules inside the kagome lattice with two neighboring adatoms (green spectra) show a pair of broad resonances between ~±1.4 and ~±2 mV (Fig. 5d). In view of the hybridization of the YSR states in the kagome precursors in Fig. 4, we assign this pair of broad resonances to YSR bands. In agreement with the observations above, molecules that are not surrounded by two Cl atoms exhibit sharp and energetically isolated states. For example, a molecule inside a pore of the kagome lattice (blue spectrum) shows a YSR state close to the coherence peaks (compared to isolated molecules in Fig. 3). Molecules at the edge of the lattice show either a YSR state around ±1.35 mV, corresponding to the Fermi energy of the sample (red spectrum), or close to the pairing energy (yellow spectrum), depending on the precise environment.

To map out the YSR bands in larger kagome domains and search for possible edge states, we show d$I$/d$V$ maps taken at various energies starting from the Fermi energy in Fig. 5e. Interestingly, we find increased intensity along the domain boundaries of the kagome lattice inside the island (indicated by blue lines). In contrast, the domain boundaries at the edge of the island do not light up. This indicates that we do not observe a discretized edge mode along the finite boundaries of the domains, but rather an edge effect. This zero-energy feature is thus related to defects in the kagome lattice, but also potentially to local changes in the ratio between Cl adatoms and FeP molecules. With increasing bias voltage, the enhanced intensity at the domain boundaries vanishes and the d$I$/d$V$ signal is seen

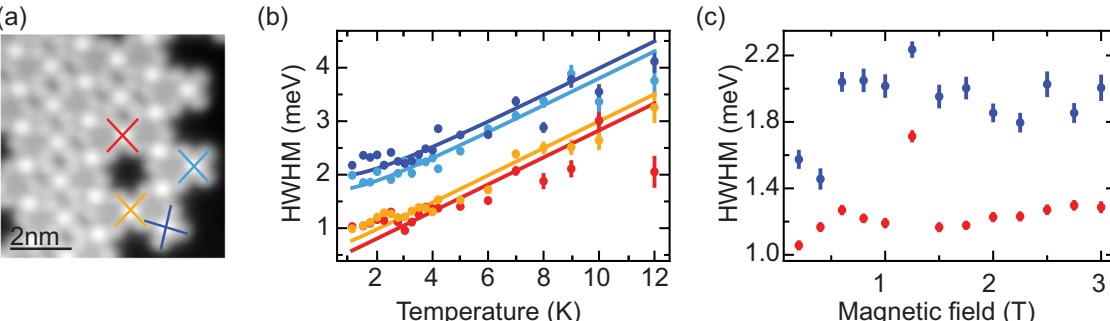

**Fig. 6 | Evolution of Kondo resonance with increasing temperature and magnetic field. a** Topographic image of an island featuring a kagome lattice in which we investigate the evolution of the d*I*/d*V* spectra with increasing temperature and magnetic field. Two molecules are within the island (marked by red and yellow crosses), and two molecules are at the edge of the domain (indicated by light and dark blue crosses). The analysis of these spectra is shown in (**b**) and (**c**). **b** The

evolution with temperature of the width of the Kondo resonances (circles—see Supplementary Note 5 for more details) for these four molecules follows that of a typical Kondo system as indicated by fits derived from ref. 40. **c** Similarly, we do not observe any variation in the resonance widths within the range of magnetic field accessible in our experiments (see Supplementary Note 6 for more details). Error bars correspond to the standard deviation of the fit parameters.

delocalized over the entire kagome domains. This is consistent with extended YSR bands.

Based on the analysis of the coupling starting from the single units (Fig. 4), the kagome lattice realized here involves one fermionic degree of freedom per site. Since we found good agreement between YSR hybridization within kagome precursors and a tight-binding model, we first consider a model in which this approach is expanded to the kagome lattice formed by YSR states induced by the FeP molecules. We assume identical hoppings along the different directions. As the YSR states are centered on a kagome lattice, the simplest tight-binding model with nearest-neighbor hopping predicts the formation of three YSR-derived kagome bands: Two dispersive bands with a linear dispersion around the *K* points, and a flat band (all of which appear twice within the superconducting gap, symmetrically about the Fermi energy). In general, the d*I*/d*V* spectra reflect the local density of states and include contributions from all parts of the Brillouin zone. Thus, they do not allow for a direct identification of the individual bands or *k*-dependent band gaps. In some cases, Fourier transforms of spatially resolved data allow for mapping the dispersion of electronic states[39]. However, in our case, the limited sizes of the domains, as well as the irregular boundaries, impede such a direct analysis. As a result, our experiments preclude direct identification of the dispersion of the YSR bands.

While our data show broadened spectral features that are consistent with YSR bands, we cannot directly correlate our data with the simple kagome band structure. In particular, we do not observe a peak indicative of a flat band. In fact, the flat band is a fine-tuned result specific to isotropic nearest-neighbor hopping and the normal state. We expect that the YSR state is induced by the $d_{z^2}$ orbital of Fe as it is most strongly coupled to the substrate. The rotational symmetry of this orbital justifies a treatment using isotropic hopping parameters. However, the slow $1/r$ decay of YSR wavefunctions may well induce longer-range hopping, and spin–orbit coupling introduces *p*-wave pairing correlations, which are neglected in the simple tight-binding model. We show in Supplementary Note 6 that both of these effects add a dispersive correction to the flat band. This broadening of the flat band makes our data (Fig. 5d) qualitatively consistent with such extended tight-binding models. We note that due to the absence of a direct bandgap in the kagome band structure, all three bands would indeed be detected as a single broad feature in d*I*/d*V*.

Finally, we investigate the magnetic fingerprints of the molecules when superconductivity in the underlying substrate is quenched. Molecules at the edge and within the islands (see Fig. 6a) all display a Kondo resonance, albeit with different widths (see Supplementary

Note 5). This is in line with the different binding energies of the YSR states indicating different coupling strengths to the substrate electrons. To reveal potential signatures of Kondo-lattice behavior we characterize the evolution of these Kondo resonances with temperature (Fig. 6b) and external magnetic field (Fig. 6c). The temperature-dependent measurements show good agreement with the universal scaling of the half width at half maximum (HWHM) of the Kondo resonance as evidenced by the fits (using ref. 40) shown as solid lines in Fig. 6b. Furthermore, we do not observe a splitting or broadening of the Kondo resonance for either molecule when applying an external magnetic field up to 3 T, thus giving a lower bound for their characteristic magnetic fields. In agreement with our earlier conclusion of small magnetic interactions of the YSR states, magnetic coupling towards Kondo-lattice behavior is also small and is not resolved in our experiments.

In conclusion, we have employed a self-assembly strategy to create a kagome lattice of spin-carrying FeP molecules and Cl atoms on a superconducting Pb(111) surface. The role of the Cl atoms is both to stabilize the triangular tiles of the kagome lattice and to shift the YSR states of the FeP deeper into the superconducting energy gap. By varying the ratio of Cl atoms and FeP molecules, we could tune the size of the kagome domains. This allowed us to track the hybridization of the YSR states and to infer the existence of extended YSR bands in the long-range ordered kagome lattice.

Magnetic coupling in the kagome lattice is weak, both compared to the YSR hybridization and to the individual Kondo coupling to the substrate. For future experiments, we suggest increasing the magnetic coupling by tuning the distance between the spins in the kagome lattice and using a 2D substrate. This may lead to Kondo-lattice or spin-liquid behavior in the normal state, and to exotic excitations within the superconducting gap.

## Methods

For all sample preparations, the Pb substrate was cleaned by sputtering with Ne[+] ions at 0.9 kV and annealing to 430 K under ultra-high vacuum (~$10^{-10}$ mbar) conditions. The molecules were subsequently evaporated from a Knudsen cell held at 485 K. The sample was then transferred into the STM chamber, where the measurements were performed at a temperature of 1.6 K. d*I*/d*V* spectra were acquired using a conventional lock-in technique, and after opening the feedback at $V_{bias}$ = 5 mV, *I* = 200 pA. The amplitude of the voltage modulation $V_{rms}$ is indicated in the figure captions. Details concerning the deconvolution of the d*I*/d*V* spectra and subsequent fit of the sample DOS are available in Supplementary Note 3.

## Data availability

The data that support the findings of this study have been deposited in the Zenodo database under accession code https://doi.org/10.5281/zenodo.10658486.

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

## Acknowledgements

We thank Nils Bogdanoff and Benjamin Heinrich for fruitful discussions. We thank Marie-Laure Bocquet for discussions of the molecular kagome structure. We acknowledge financial support from the Deutsche Forschungsgemeinschaft (DFG, German Research Foundation) through projects 277101999 (CRC 183, project C03) and FR2726/10-1.

## Author contributions

L.F. carried out the experiments with the assistance of G.R. K.J.F. guided the experiments. F.v.O. did the theoretical model. All authors discussed the results. L.F., F.v.O. and K.J.F. wrote the manuscript with input from G.R.

## Funding

## Competing interests

The authors declare no competing interests.
