## [Peer Review File · Nature Communications]

REVIEWER COMMENTS

Reviewer #1 (Remarks to the Author):

The authors report a low-temperature (1.6 K) scanning tunnelling microscopy (STM) and spectroscopy (STS) study on a two-dimensional (2D) kagome lattice consisting of magnetic iron-porphine (FeP) molecules and chlorine (Cl) atoms on a superconducting Pb(111) surface. The main result of the manuscript seems to be the observation of Yu-Shiba-Rusinov (YSR) states - spin-polarized bound states resulting from the exchange interaction between the FeP magnetic moments and Cooper pairs of the Pb(111) superconductor – and hybridization of the latter via electron hopping between nearest-neighbour kagome-arranged FeP molecules. The authors claim these hybrid states resulting from hybridisation of YSR states give rise to electronics bands protected by the superconducting gap.

The study is timely, given the significant research interest there is currently in interactions of magnetic moments with superconductors, and in quantum phases emerging from many-body interactions in solid-state systems with flat electronic bands. Therefore, I do believe that the manuscript deserves to be published in Nature Communications, provided that the following comments are addressed in a following version of the draft:

- The claim of hybridisation between YSR states of nearest-neighbour FeP molecules seems convincing, in particular looking at Fig. 4 and related discussion. However, the formation of YSR bands, with a continuous energy spectrum within these bands, does not seem obvious. In other words, hybridisation between a finite number of states of adjacent molecules is not a synonym of continuous energy bands. What exact features in Fig. 5d are related to such claimed YSR bands? Can such features be explained quantitatively by the hybridisation of YSR states positioned in a kagome geometry, and if so, how? The claim of emergence of YSR bands is central to the manuscript, so it would be good that the authors explain such emergence in detail and if possible quantitatively.

- The authors seem to imply that that the kagome arrangement of the FeP molecules is important for the emergence of YSR bands and related dI/dV features in Fig. 5d. However, this doesn't seem to be explained explicitly nor in detail. In other words, what is the effect of the kagome arrangement on the claimed YSR bands? If the FeP molecules were arranged in a periodic geometry other than a kagome lattice, how would that affect the dI/dV signatures of the claimed YSR bands?

- What is the relationship between the intrinsic kagome band structure in Fig. 1a (with the Dirac and flat bands), the claimed YSR bands, the dI/dV features in Fig. 5d resulting from hybridisation of YSR

states, and the band structure resulting from the Hamiltonian in the SM Section S6? Can parameters Δ and t in the Hamiltonian in the SM Section S6 be estimated from the experimental dI/dV features in Fig. 5d? What is the band structure resulting from the Hamiltonian in the SM Section S6, and how does it compare to the intrinsic kagome band structure in Fig. 1a?

- What is the nature of the bonding between FeP molecules and Cl atoms? What is the electronic configuration of the Cl and Fe atoms within the kagome assembly? Is the strength of the FeP-Cl bonding consistent with the implied tight-binding hopping term t (in the Hamiltonian in the SM Section S6) between nearest-neighbour FeP molecules?

- There seems to be some FeP molecules trapped in the pores of the kagome arrangement, but this is not discussed explicitly in the manuscript. Do these trapped FeP molecules have an effect on the dI/dV signatures of adjacent kagome-arranged FeP molecules? How do the dI/dV spectra look for these trapped molecules?

Reviewer #2 (Remarks to the Author):

The manuscript by Farinacci and coworkers reports on Yu-Shiba-Rusinov bands in a self-assembled kagome lattice, which was done by scanning tunneling microscopy. The manuscript is mostly experimental reporting in detail about the preparation process followed by a discussion of the experimental observations with very little theory support. While the overall idea looks interesting, I believe that the analysis is inconsistent and the overall conclusions could be more substantive. Furthermore, I am not even sure if the lattice is actually a kagome lattice. Based on the overall assessment and the more detailed points below, I cannot recommend publication of this manuscript.

1) page 3, right column: the authors claim that tip-approach measurements reveal that the system is in a screened spin ground state, which is detailed in section S4 of the SM. In the SM, the authors cite Ref. [1], which is their own work, where they conclude that the exchange coupling weakens, when the tip approaches. In Ref. [5], the authors state that the exchange coupling increases when the tip is approached. How do the authors resolve this inconsistency?

2) page 3, right column: the authors write, "This is in accordance with measurements taken in an external magnetic field, which quenches superconductivity in the Pb substrate and reveals the presence of a Kondo resonance (Fig. 3d)." Does this mean that the presence of the Kondo peak shows that the system is in the screened spin regime? This would not make sense, since in Ref. [30],

the Kondo peak was observed in both regimes. So, how is their statement to be understood? Can the authors explain their statement in more detail?

3) figure 3b, the authors write that the YSR state at ± 1.87 mV in the black curve originates from the molecule at the red dot. How do they conclude that? This should be visible in Fig. 3c and d, but I only see dark. Can this region be enhanced to see that the origin is from the red dot molecule?

4) page 4, left column, the authors are talking about "local variations of the electronic density of states at the Fermi level." Do they mean the changes in YSR energies as observed in Ref. [30]? How are such variations taken into account in the subsequent analysis? Why would such local density of states variations only affect molecules at the edge?

5) figure 4c,d, this does not really look like a bonding and anti-bonding example. Each molecule seems to have a center feature surrounded by four other features. In both c and d, the molecules look like that they are sharing one of the outer features (they do look they have different intensity though), which does not look like an anti-bonding example. The authors mention a tight-binding model (the sketches below c and d), but they never give any details about this. Is this on the level of the hydrogen molecule or is this related to S6 in the SM?

6) page 5, right column, the authors talk about the RKKY interaction (which was never defined). Although the authors exclude RKKY coupling, I wonder why they considered it at all. The RKKY interaction relies on conduction electrons, but in a superconductor, the charge carriers have zero spin (at least in an s-wave superconductor), so how would that work?

7) Throughout the manuscript, the authors make a strong point that the Cl atoms play a vital role in the formation of this lattice, in particular, the hybridization. However, they ignore the presence of the Cl atoms in the characterization of the kagome lattice. Including the Cl atoms would turn the structure into a honeycomb lattice or maybe a honeycomb-kagome lattice. A honeycomb lattice would not exhibit a flat band, which the authors also say they did not observe. Can the authors reconcile this?

8) Along the lines of 7), the authors write "While our experiments thus preclude the identification of a flat band, we also note that spin-orbit coupling introduces p-wave superconducting pairing which adds a dispersive correction to the

at bands (see section S6 in supplementary material)." I would like to see a more detailed explanation of this statement in the main text.

9) page 7, left column, the authors write "In general, the dI/dV spectra include contributions from all parts of the Brillouin zone and thus do not allow for a direct identification of the individual bands or k -dependent band gaps." I do not agree with this statement, since the information about the k -vector is contained in the **local** density of states, which is measured in the STM. This is how the analysis could be done in Ref. [38]. What is different in this manuscript? Even if the Fourier transform is difficult to do, some modulation of the peak intensities at different lattice sites should allow for the indication of a k -vector. This would be a much more convincing argument for YSR bands compared to what the authors offer.

Reviewer #3 (Remarks to the Author):

Laetitia Farinacci et al. reported on a self-assembled kagome lattice of FeP-Cl molecules on SC Pb(111) substrate, where the temperature-dependent de-chlorination was observed and used as a controlling factor, and YSR states were observed and studied by dI/dV maps. The authors tracked the hybridization of the YSR states by tuning the kagome domain sizes, and finally studied the Kondo resonance by temperature- and magnetic-field-dependent measurements after quenching the substrate's superconductivity in fields. Unfortunately, the authors claimed that flat bands as a typical feature in kagome lattice were not possible to be experimentally observed due to the limitations of the domain size and the irregular domain boundary conditions.

Overall, this manuscript is well-written and scientifically sound. However, I would like to raise the following minor questions concerning this manuscript:

1. I suggest that the authors add the pristine molecular structural formula of FeP-Cl in Figure 1 as inset or somewhere else in the figures just to make it clearer and more readable.
2. What is the structure (or self-assemble structure) upon deposition when the substrate is held at even lower temperatures (< 230 K), where de-chlorination is not triggered? Does the pristine FeP-Cl self-assemble on Pb? If so, how is its electronic properties different compared to de-chlorinated FeP and Cl-atom kagome lattice?
3. The kagome lattice pores seem to different topographic contrast, e.g., in the yellow unit cell area in Fig. 2c, three pores are grey while one pore pitch black, does this indicate different electronic states in the pores and why?

4. In Fig. 3b, the dI/dV on the central molecule (red spectrum) seems to have a huge dip at around 2 mV which is below zero, is this the sign for a negative differential resistance? If so, why is there NDR around ± 2 mV in the red spectrum?

5. Is there any DFT calculation results for the exchange coupling energies in this kagome lattice? How much meV are the RKKY and YSR coupling terms? What about the molecular geometrical configurations and electronical charge mapping? Is there any chemical bonding forming after annealing the sample? If so, how does the bonding affect the exchange coupling strength?

In summary, I found the results in this manuscript solid and well-structured, but at the same time not very surprising. The key highlight is the size control of the kagome domains through ratio control of Cl/FeP by temperature, the main findings of the manuscript, i.e., the evolvement of YSR states inside the SC Pb gap, are well-expected. The fact that the flat bands cannot be experimentally observed is quite a pity, which could be rather interesting using this size-variable FeP-Cl kagome lattice as a template. Considering the broad interest of readership and high prestige of Nature Communications, I personally do not feel this manuscript is fit for publication in Nature Communications. The ACS Nano could be a better venue for this manuscript. Nonetheless, I will be happy to read the revised manuscript after the authors resolve the above-mentioned issues.

Reviewer #1 (Remarks to the Author):

The authors report a low-temperature (1.6 K) scanning tunnelling microscopy (STM) and spectroscopy (STS) study on a two-dimensional (2D) kagome lattice consisting of magnetic iron-porphine (FeP) molecules and chlorine (Cl) atoms on a superconducting Pb(111) surface. The main result of the manuscript seems to be the observation of Yu-Shiba-Rusinov (YSR) states - spin-polarized bound states resulting from the exchange interaction between the FeP magnetic moments and Cooper pairs of the Pb(111) superconductor – and hybridization of the latter via electron hopping between nearest-neighbour kagome-arranged FeP molecules. The authors claim these hybrid states resulting from hybridisation of YSR states give rise to electronics bands protected by the superconducting gap.

The study is timely, given the significant research interest there is currently in interactions of magnetic moments with superconductors, and in quantum phases emerging from many-body interactions in solid-state systems with flat electronic bands. Therefore, I do believe that the manuscript deserves to be published in Nature Communications, provided that the following comments are addressed in a following version of the draft:

We thank the referee for their report and positive assessment of our work. We respond to the individual points below and amend the manuscript accordingly. Changes to the manuscript and supplementary material are marked in red in separate files.

- The claim of hybridisation between YSR states of nearest-neighbour FeP molecules seems convincing, in particular looking at Fig. 4 and related discussion. However, the formation of YSR bands, with a continuous energy spectrum within these bands, does not seem obvious. In other words, hybridisation between a finite number of states of adjacent molecules is not a synonym of continuous energy bands. What exact features in Fig. 5d are related to such claimed YSR bands? Can such features be explained quantitatively by the hybridisation of YSR states positioned in a kagome geometry, and if so, how? The claim of emergence of YSR bands is central to the manuscript, so it would be good that the authors explain such emergence in detail and if possible quantitatively.

We agree with the Referee that the distinction between discrete states and continuous bands is delicate. Strictly speaking, electronic bands only exist in infinite (or periodic) systems. However, band formation is ultimately due to hybridization of the states of the individual building blocks. Thus, it can also be tracked by building larger structures by repeatedly adding building blocks. In practice, one can use the concept of bands when both energy and momentum are sufficiently well defined. By the Heisenberg uncertainty relation, the momentum of a state is well-defined up to the inverse of the size of the domains. When this uncertainty is small compared to the size of the Brillouin zone, band theory becomes useful to describe the physics of the system. Similarly, the energy uncertainty needs to be small compared to the bandwidth.

This is the rationale behind the approach that we take in our experiments. Starting from a single building block with a discrete and localized state, we increase the number of building blocks and track the evolution of the states. We observe hybridization as manifested in a splitting and delocalization of the states. In small structures, we can identify individual hybridized states. This implies that our energy resolution is better than the bandwidth. In larger structures, the states start to overlap. In Fig. 5d, we show dI/dV spectra, where we no longer resolve individual YSR states. While this alone would not necessarily signify band formation, our systematic study starting from individual building blocks and

showing hybridization in larger structures is strong indication of band formation. It is for this reason that we employ the concept of YSR bands to describe the hybridization of YSR states in large domains across 2D islands. One limitation inherent in our STM measurements is, of course, that they do not give direct access to the momentum of an electronic state (as we discuss below in response to the second question).

We would like to mention that as a matter of principle, the concept of bands is always an idealization given that real solids are finite (and subject to interactions). If one were to probe the solid with sufficient energy and/or momentum resolution, one would discover that energies are discrete and momentum is not a good quantum number. Nevertheless, the concept of bands is extremely useful over a wide range of energy and momentum scales.

- The authors seem to imply that that the kagome arrangement of the FeP molecules is important for the emergence of YSR bands and related dI/dV features in Fig. 5d. However, this doesn't seem to be explained explicitly nor in detail. In other words, what is the effect of the kagome arrangement on the claimed YSR bands? If the FeP molecules were arranged in a periodic geometry other than a kagome lattice, how would that affect the dI/dV signatures of the claimed YSR bands?

In general, all lattices lead to YSR band formation if the YSR states hybridize sufficiently strongly. However, the kagome arrangement of molecules is particularly interesting as kagome lattices exhibit an intriguing band structure featuring both a flat band and strongly dispersing bands with Dirac character. (This assumes a single orbital per site, nearest-neighbor hopping, and identical hopping between all nearest neighbors.) Hence, we were hunting for these characteristics in our experiments. To do so, we grew a kagome structure hosting YSR states by self-assembly – a non-trivial task by itself. While it is straightforward to deduce the structure of the lattice by STM, it is more challenging to obtain information concerning the electronic band structure. We agree that the discussion on the expected features and the related challenges fell short in the previous version of the manuscript. We have now expanded on this point in the new version.

In brief, the hallmarks of the kagome band structure (with identical hopping between sites) are strongly dispersing bands and a flat band. While STM experiments cannot access the band structure directly, they measure the local density of states. Spatially averaged it consists of a broad feature (reflecting the dispersive band) limited by van Hove singularities and a sharper peak reflecting the flat band, as shown in Figure 1 below.

Figure 1 Density of states of the kagome lattice. The energy scale is units of the hopping parameter t . The von Hove singularities of the flat bands are seen as central peaks and the flat band at higher energies. Note that these features are expected on the energy scale of the width of the peaks in dI/dV within the superconducting energy gap, i.e. on a scale of 0.6meV .

In Fig. 5d, we show dI/dV spectra of several molecules within the kagome lattice. Taking advantage of the atomic resolution of STS, we identify which molecules form part of the lattice. The molecules which are part of the kagome lattice exhibit broad features, thus resembling bands. In contrast to simple expectations, we do not find an additional sharper peak which would be a fingerprint of the flat band.

We have now extended our discussion of possible reasons why the signature of the flat band is not apparent. First, one should exercise caution when interpreting these data: (i) We investigate YSR bands within a very small energy scale of $\sim 0.6\text{meV}$. This renders the identification of the band structure quite challenging. (ii) While STS is a great tool to track the evolution of the states starting with the individual building blocks, it is much more challenging to obtain information on momentum space and the total density of states. Since STS measures the *local* density of states, it exhibits spectroscopic variations between sites of a lattice due to scattering at domain boundaries such as those observed in Fig. 5d. Moreover, the theoretical prediction of flat bands makes several assumptions, which may not apply to the experimental system. The slow decay of the YSR states suggests that there is substantial next-nearest-neighbor coupling. Moreover, the band structure is a prediction for normal-metal bands. Given that YSR bands form within the superconducting gap, their formation may not only involve hopping between neighboring sites, but also pairing interactions. We have now added more details, including an illustrative figure, in Sec. S6.

We had previously shown the band structure in Fig.1a, but as it seems to have led to confusion, we have removed it in the current version.

- What is the relationship between the intrinsic kagome band structure in Fig. 1a (with the Dirac and flat bands), the claimed YSR bands, the dI/dV features in Fig. 5d resulting from hybridisation of YSR states, and the band structure resulting from the Hamiltonian in the SM Section S6? Can parameters Δ and t in the Hamiltonian in the SM Section S6 be estimated from the experimental dI/dV

features in Fig. 5d? What is the band structure resulting from the Hamiltonian in the SM Section S6, and how does it compare to the intrinsic kagome band structure in Fig. 1a?

This question is related to the previous one, where we have explained the relation of the total density of states and the kagome band structure as well as the challenge of comparing the total DoS with the local DoS. Additionally, we note that the band structure discussed so far results from a tight-binding model with nearest-neighbor coupling. We consider this model first as it is the simplest approach to describe the physics at play in a kagome lattice and because it reproduces the hybridization between YSR states in kagome precursors very well, as shown in Fig.4. We can estimate the hopping term from the YSR state energy splitting: $t \approx -0.16 meV$.

A main result of this tight-binding model is the appearance of a flat band, which we cannot clearly identify in our experiments (as already discussed in response to the previous question). While experimental limitations are at play we also expect that perturbations to the tight-binding model induce dispersion in this band.

As discussed in our response to the previous question, the simple Kagome band structure relies on several assumptions, only some of which we can test experimentally. Importantly, the tight-binding approach uses identical nearest-neighbor hopping amplitudes. With the YSR states originating from the d -orbitals of the FeP molecules, this is not necessarily the case. However, we argue that this is a reasonable assumption. As for other molecules such as Mn phthalocyanine, we expect the YSR state to be induced by the d_{z^2} -orbital which is most strongly coupled to the substrate. The rotational symmetry of this orbital may justify a description using identical hopping amplitudes to neighboring sites. Perturbations by the underlying substrate are also expected to be small as we do not observe differences in hybridization strength for dimer structures which are oriented differently on the surface.

In general, couplings beyond nearest neighbors and pairing interactions alter the band structure and will broaden the flat band. The algebraic decay of YSR states (on scales small compared to the superconducting coherence length) is likely to make hybridization long range. The longer-range couplings can only be estimated from data on larger structures, where our energy resolution no longer suffices to resolve individual levels. (We have checked that the data on the trimer are consistent with substantial levels of next-nearest-neighbor hopping.)

Moreover, in addition to hybridization, the formation of YSR bands may involve pairing interactions. Superconductivity induces a duplication of YSR resonances symmetric about the Fermi energy. Superconducting correlations couple these pairs of bands, with a coupling strength that is in general k -dependent and possibly long ranged. An upper limit of the pairing Δ_p would be the gap of the bare Pb substrate, which is $1.35 meV$. However, the actual value of Δ_p is difficult to estimate, as it depends on the spin-orbit coupling of the substrate. (Note that here, Δ_p does not refer to the gap of the substrate superconductor, but the pairing interactions involving *spin-polarized* YSR states.)

- What is the nature of the bonding between FeP molecules and Cl atoms? What is the electronic configuration of the Cl and Fe atoms within the kagome assembly? Is the strength of the FeP-Cl bonding consistent with the implied tight-binding hopping term t (in the Hamiltonian in the SM Section S6) between nearest-neighbour FeP molecules?

The molecules inside the kagome lattice which are surrounded by Cl adatoms have similar appearance and spectroscopic signatures as those obtained on preparations where no Cl remains on the sample (Fig.2g and h). Thus, we do not expect strong hybridization between the Cl and FeP orbitals. Most likely, the lattice is stabilized by hydrogen bonds as Cl is very electronegative. This interpretation is further corroborated by the fact that the YSR states of molecules surrounded by Cl atoms shift deeper into the superconducting gap than those of isolated molecules. This is consistent with a “local gating” effect of the Cl atoms on the YSR states.

The hopping amplitude t of the tight-binding model can be estimated from the YSR splitting in Fig.2 ($t \approx -0.16\text{meV}$) but does not relate to the FeP-Cl bonding strength. In contrast, YSR hybridization is dominated by the overlap of the YSR wave functions. These wave functions are localized in the superconducting substrate and distinct from the orbitals of the magnetic impurities which induce the YSR states. In particular, as shown in [20], the YSR states have a much larger spatial extent.

- There seems to be some FeP molecules trapped in the pores of the kagome arrangement, but this is not discussed explicitly in the manuscript. Do these trapped FeP molecules have an effect on the dI/dV signatures of adjacent kagome-arranged FeP molecules? How do the dI/dV spectra look for these trapped molecules?

The molecules trapped inside the pores of the Kagome lattice do not interact or affect the molecules inside the lattice. They display a sharp YSR state close to the gap edge. This is shown in the dark blue spectrum of Fig.5d. We now make this more explicit in the text.

Reviewer #2 (Remarks to the Author):

The manuscript by Farinacci and coworkers reports on Yu-Shiba-Rusinov bands in a self-assembled kagome lattice, which was done by scanning tunneling microscopy. The manuscript is mostly experimental reporting in detail about the preparation process followed by a discussion of the experimental observations with very little theory support. While the overall idea looks interesting, I believe that the analysis is inconsistent and the overall conclusions could be more substantive. Furthermore, I am not even sure if the lattice is actually a kagome lattice. Based on the overall assessment and the more detailed points below, I cannot recommend publication of this manuscript.

We thank the referee for reading our manuscript and their critical comments. However, we disagree with their judgement on inconsistent conclusions. In the following we respond to their criticism which should remove all misunderstandings of inconsistency. Changes to the manuscript and supplementary material are marked in red in separate files.

1) page 3, right column: the authors claim that tip-approach measurements reveal that the system is in a screened spin ground state, which is detailed in section S4 of the SM. In the SM, the authors cite Ref. [1], which is their own work, where they conclude that the exchange coupling weakens, when the tip approaches. In Ref. [5], the authors state that the exchange coupling increases when the tip is approached. How do the authors resolve this inconsistency?

Both papers cited by the referee are correct and entirely consistent with each other. Both papers exploit that the interaction between adsorbed (magnetic) molecules and an STM tip can be used to modulate the strength of the exchange coupling between the magnetic impurity and the superconducting substrate. The nature of the interaction between molecule and STM tip depends sensitively on their distance. At large tip-molecule distances, the interaction is dominated by van der Waals forces, which attract the molecule to the tip. As a result, the interaction increases the distance

between substrate and molecule, thereby *decreasing* the exchange coupling between molecular spin and substrate electrons. At shorter tip-molecule distances, the tip-molecule interaction is dominated by direct Pauli repulsion and pushes the molecule back toward the surface. This leads to an increase of the exchange coupling. In Ref. [1], we actually observed both regimes. As the tip approaches the molecule, we first observe a shift of the YSR state towards the Fermi energy. This is the behavior expected for attractive tip-molecule coupling and a decrease in the exchange coupling. As the tip approaches the molecule further, we observe that the shift of the YSR state reverses direction. This is exactly what is expected once the tip-molecule force changes from attractive to repulsive and the exchange coupling begins to increase. Ref. [5] is based on the same physics but deals with a different system. For this system, the authors do not observe a YSR state when the tip is far from the molecule. The exchange coupling is thus very weak in the native state (without STM tip). When approaching the tip, the exchange coupling is initially further weakened due to the molecule-tip attraction. However, this will not lead to an observable effect (as there are no YSR states in the first place). Only when the tip approaches the molecule closely – almost forming a point contact between tip and molecule – the authors observe the emergence of a YSR state inside the superconducting gap. At this point, the Pauli repulsion between tip and molecule increases the exchange coupling to the point that the YSR state forms and becomes observable. Both studies are thus entirely consistent with each other.

2) page 3, right column: the authors write, "This is in accordance with measurements taken in an external magnetic field, which quenches superconductivity in the Pb substrate and reveals the presence of a Kondo resonance (Fig. 3d)." Does this mean that the presence of the Kondo peak shows that the system is in the screened spin regime? This would not make sense, since in Ref. [30], the Kondo peak was observed in both regimes. So, how is their statement to be understood? Can the authors explain their statement in more detail?

We agree with the referee that the quoted sentence may have been prone to misinterpretation. We deduce the screened-spin nature from the tip-approach measurements as explained in the supplement, and not from the existence of the Kondo resonance. At the same time, the existence of the Kondo resonance remains consistent with this fact and supports the statement that the molecules constitute quantum spins. Indeed, a Kondo resonance is a direct manifestation of the quantum spin nature.

The transition between the free-spin to the screened-spin regimes occurs when the Kondo temperature increases to exceed the pairing gap. Thus, as stated by the referee, there is a Kondo resonance in the screened spin as well as in the free spin regimes once superconductivity is quenched. In fact, the universal scaling of the energy of the YSR states with the strength of the Kondo resonance was probed experimentally by one of us in Reference [30]. To further strengthen the comparison between our measurements in the normal and superconducting states, we have now explicitly extracted the Kondo temperature: $T_K = 7.9 \pm 0.3$ K. (The error bar corresponds to the standard deviation of the fit to a Fano-Hurwitz function [Turco2023].) Comparing it to the superconducting gap of the sample, we find that $k_B T_K = 0.68$ meV. This is beyond the quantum phase transition, which is expected to take place for $k_B T_K \approx 0.3 \Delta_{Pb}$ [Satori1992], giving further support to our claim that the system being in the strong coupling regime.

The new determination of the Kondo temperature is based on a recent paper using a new fit function, coined Hurwitz function, that accurately takes into account the broadening of the tip DOS due to temperature [Turco2023]. We have therefore amended the fits in Fig.3d and Fig.6 of the main text as well as in Fig.S5 and Fig.S6. The conclusions of our paper remain unaffected by this change.

[Turco2023] Turco et al., arXiv:2310.0932

[Satori1992] Satori et al., J. Phys. Soc. Jpn. 61, 3239 (1992)

3) figure 3b, the authors write that the YSR state at $\pm 1.87\text{mV}$ in the black curve originates from the molecule at the red dot. How do they conclude that? This should be visible in Fig. 3c and d, but I only see dark. Can this region be enhanced to see that the origin is from the red dot molecule?

The red spectrum taken at the center of the central molecule shows a strong YSR resonance with large asymmetry. This is also reflected by the maps taken at positive and negative bias voltage. Additionally, we observe intensity of the same state on the ligands owing to cotunneling through higher molecular states, thus mapping out the molecular orbitals [Ref. 35 in the previous version, now Ref. 36]. The spectra on the other molecules at the corner of the structure show a faint signal at the same energy and – due to this small intensity – cannot be seen in the maps. Indeed, these maps show that the signal is centered around the molecule marked by the red dot. The pair of resonances at $\pm 1.87\text{meV}$ is thus due to the presence of a YSR state originating from the central molecule but extending in space. Hence, it is only faintly detected above the neighboring molecules, both in dI/dV spectra and dI/dV maps.

Figure 1 dI/dV map at -1.87 mV

Figure 2 dI/dV map at $+1.87\text{ mV}$

Here, we show the dI/dV maps of Fig.3 using a color scale that enhances the dark regions. These maps show where the signal is localized. Importantly we see that the signal is not localized around the

molecules at the edge of the structure. The signal there is faint. This is in accordance with the dI/dV spectra. These are a measure of the same quantity but at a specific position and show that the resonances have very weak intensity above the molecules at the edge of the structure.

4) page 4, left column, the authors are talking about "local variations of the electronic density of states at the Fermi level." Do they mean the changes in YSR energies as observed in Ref. [30]? How are such variations taken into account in the subsequent analysis? Why would such local density of states variations only affect molecules at the edge?

We mean that the YSR states shift in energy due to a local potential. This has been previously observed for molecular assemblies in Ref. [32] and for atoms influenced by a charge density wave in Ref. [36] (Ref. [38] in the new version of the manuscript). Reference [30] also reported a shift of YSR states associated with a moiré structure modulating the exchange coupling to the substrate.

In more detail, in Ref. [32], different molecular assemblies were constructed, and it was shown that YSR states emerge at different energies depending on the presence of neighboring molecules. In Ref. [36], the charge density wave of the $NbSe_2$ substrate spatially modulates the charge density. This translates into a variation of the YSR energy of magnetic adatoms as a function of location. Very similar to these cases, we observe in Fig.3 that FeP molecules display YSR states at different energies depending on their local surroundings. When only one Cl adatom is in their vicinity, they display a YSR state close to the gap edge. When two Cl adatoms are in their vicinity, they display a YSR state well inside the gap. These observations can be rationalized by the electronegative nature of the Cl atoms. This makes the Cl atoms act as a local gate which modifies the electron density in the vicinity. Thus, different molecular surroundings (one vs. two Cl adatoms) lead to different energies of the YSR state (at the gap edge vs. inside the gap). In our subsequent analysis, we focus on molecules that have identical surroundings (two Cl adatoms). As shown in Fig.4, it is in this case that we observe YSR hybridization.

All molecules inside the kagome lattice have identical surroundings. Therefore, we do not expect variations of the density of states at the Fermi level between them. Our observations are thus in good agreement with earlier publications attributing the shift to local changes in the atomic-scale surroundings.

5) figure 4c,d, this does not really look like a bonding and anti-bonding example. Each molecule seems to have a center feature surrounded by four other features. In both c and d, the molecules look like that they are sharing one of the outer features (they do look they have different intensity though), which does not look like an anti-bonding example. The authors mention a tight-binding model (the sketches below c and d), but they never give any details about this. Is this on the level of the hydrogen molecule or is this related to S6 in the SM?

As pointed out by the referee, it is important to understand that the shape of the YSR states of the individual molecules is already rather complex. As a result of the molecular adlayer, one does not directly image the YSR states present in the superconducting substrate. Instead, the observed "shape of the YSR states" arises from co-tunneling through non-magnetic molecular states, essentially mapping out the molecular orbital structure [35]. To understand the hybridization of YSR states in the substrate, we thus focus on deviations of the intensity distributions from the original shapes when two or more molecules interact with each other. In doing so, we keep in mind that all hybrid YSR states are again seen via co-tunneling through the molecular states. Evidence of hybridization comes from the

splitting of resonances in dimer structures. More specifically, in Fig. 4c,d the molecules display two YSR states with distinct intensity distributions. One state (named alpha, Fig. 4c) exhibits increased intensity localized between the molecular centers – described by the referee as “sharing the outer features”. If this was a simple overlap of intensities of the molecular states, there should not have been a split into two resonances and no reduction of intensity at the other corners. While the alpha state shows a clearly enhanced intensity between the two molecules – consistent with a symmetric superposition of YSR states – the nodal plane for the beta state is less visible but clearly there is less intensity at the corner-sharing features compared to the other three lobes. As mentioned above, the dI/dV signal does not directly map the YSR wave function but rather results from the shape of the non-magnetic molecular orbitals. This effectively complicates the direct detection of nodal planes.

Coupling between two degenerate states leads to the formation of symmetric and anti-symmetric combinations of the original states – independent of the exact model used to parametrize the hybridization. The spectra are thus direct evidence of hybridization – the main conclusion that we draw and consistently expand on for the larger structures. The demonstration of hybridization is further strengthened by our analysis of a three-site chains, where the agreement is very clear as hybridization leads to different intensities of the states on the three sites.

For a basic understanding of the system, we focus on the degenerate states (measuring energy from the YSR energy of the uncoupled impurity spin) and use a tight-binding model with nearest-neighbor coupling. For a two-site chain, the Hamiltonian is

$$\begin{pmatrix} 0 & t \\ t & 0 \end{pmatrix}$$

Similarly, for a three-site chain it is

$$\begin{pmatrix} 0 & t & 0 \\ t & 0 & t \\ 0 & t & 0 \end{pmatrix}$$

A description at the level of the hydrogen molecule would be slightly more elaborate as it involves the overlap, Coulomb and exchange integrals. Due to the increased number of parameters, this model also describes our experiments for a wide range of parameters. Since it does not contribute further insights into the physics at play, we prefer to use the minimal tight-binding approach.

For illustrative purposes, we go beyond this model in the section S6 of the Supplement. Our intention here is to show possible limitations of the model, although experiments may not be able to resolve these signatures at the present energy resolution and with the available structures.

6) page 5, right column, the authors talk about the RKKY interaction (which was never defined). Although the authors exclude RKKY coupling, I wonder why they considered it at all. The RKKY interaction relies on conduction electrons, but in a superconductor, the charge carriers have zero spin (at least in an s-wave superconductor), so how would that work?

As is known since the pioneering work by P. W. Anderson and H. Suhl in 1959 [Anderson1959], Ruderman-Kittel-Kasuya-Yosida (RKKY) coupling also exists in superconductors. Superconductivity induces only a small gap around the Fermi level, leaving many electrons able to screen. In short, RKKY coupling occurs on length scales, which are small compared to the superconducting coherence length. This implies that it is due to electronic states, which are far from the Fermi energy on the scale of the superconducting gap and thus approximately unaffected by superconductivity.

The effect of RKKY interactions on the coupling of YSR states has been discussed in several experimental works [Kezilebieke2018, Ding2021, Kuester2021]. Seminal theoretical discussions of RKKY coupling in the context of YSR systems can be found in [Yao2014,Yao2015]. Given the fundamental nature of the RKKY interaction, we comment on its potential role in our system.

We thank the referee for pointing out the missing definition of the abbreviation RKKY. We have now added the definition to the manuscript.

[Anderson1959] Anderson and Suhl, Phys. Rev. 116 (1959) 898

[Kezilebieke2018] Kezilebieke et al., Nano Lett. 18 (2018) 2311

[Kuester2021] Kuester et al., Nat. Comm. 12 (2021) 6722

[Ding2021] Ding et al., PNAS 118 (2021) e2024837118

[Yao2014] Yao et al., Phys. Rev. Lett. 113 (2014) 087202

[Yao2015] Yao et al., Phys. Rev. B 90 (2015) 241108(R)

7) Throughout the manuscript, the authors make a strong point that the Cl atoms play a vital role in the formation of this lattice, in particular, the hybridization. However, they ignore the presence of the Cl atoms in the characterization of the kagome lattice. Including the Cl atoms would turn the structure into a honeycomb lattice or maybe a honeycomb-kagome lattice. A honeycomb lattice would not exhibit a flat band, which the authors also say they did not observe. Can the authors reconcile this?

The Cl atoms are instrumental for the formation of the kagome lattice of the FeP molecules. But they do not carry a spin and, consequently, no YSR states. The YSR states arise exclusively from the FeP molecules and thus form a kagome lattice. It is the hybridization of this kagome lattice of YSR states that we are ultimately interested in in this manuscript. The honeycomb-kagome lattice obtained when including the Cl atoms has essentially no direct relevance in this context.

Of course, the Cl atoms are important in that they influence the YSR states of the FeP molecules as we also explained in response to question 4. We now clarify in the manuscript that the Cl atoms do not themselves induce YSR states.

8) Along the lines of 7), the authors write "While our experiments thus preclude the identification of a flat band, we also note that spin-orbit coupling introduces p-wave superconducting pairing which adds a dispersive correction to the at bands (see section S6 in supplementary material)." I would like to see a more detailed explanation of this statement in the main text.

In the new version, the main text contains an expanded discussion of effects that may lead to dispersive corrections to the flat band. The existence of a flat band is a property of tight-binding models with isotropic nearest-neighbor hopping describing the normal state. A Kagome lattice of YSR states can differ from this in three ways. First, the orbital structure of the YSR states can be such that the hopping is no longer isotropic. Second, hopping can be longer ranged, which adds a nonzero dispersion to the flat band. And third, even if hopping is isotropic, hybridization of YSR states can be affected by pairing correlations. Essentially, this is due to processes in which a Cooper pair breaks up and the two electrons virtually occupy neighboring YSR states. (This has been widely discussed in the context of realizations of topological superconductivity in chains of magnetic adatoms.) Due to the spin polarization of the YSR state and the spin-singlet nature of Cooper pairs in the simplest models, this

process may require spin-orbit coupling. When present, this additional hybridization specific to superconductors modifies the band structure and introduces a dispersion of the flat band.

We have also extended the discussion of these effects in Sec. S6 of SM, including a new figure illustrating the effects of next-nearest-neighbor hopping and superconducting pairing on the band structure.

9) page 7, left column, the authors write "In general, the dI/dV spectra include contributions from all parts of the Brillouin zone and thus do not allow for a direct identification of the individual bands or k -dependent band gaps." I do not agree with this statement, since the information about the k -vector is contained in the **local** density of states, which is measured in the STM. This is how the analysis could be done in Ref. [38]. What is different in this manuscript? Even if the Fourier transform is difficult to do, some modulation of the peak intensities at different lattice sites should allow for the indication of a k -vector. This would be a much more convincing argument for YSR bands compared to what the authors offer.

Indeed, dI/dV measurements contain information on the local density of states which effectively integrates over k states due to the locality. Thus, electrons from all parts of the Brillouin zone contribute to tunneling (though possibly with different decay constants depending on k_{\parallel}).

We agree with the reviewer that in special cases, it is possible to obtain momentum information from STM measurements. This relies on transforming real-space information to reciprocal space by Fourier transforming. This is typically what is done in quasi-particle interference (QPI) measurements. These measurements typically require impurities as well as large dI/dV maps to ensure good resolution in reciprocal space. For YSR states, the authors of Ref. [38] use a closely related approach. Investigating finite 1D chains, their ends can be thought of as analogs of the impurities and induce standing-wave patterns. These essentially correspond to 1D particle in a box YSR states to which one can in principle assign a momentum value.

While such an analysis would be beneficial to identify the dispersion of the YSR bands in our system, several challenges impede it. First, as discussed above, the measurements of the YSR states are influenced by co-tunneling through molecular orbitals and do not directly access the YSR states in the superconducting substrate. Second, the shape of the domains is rather irregular which prevents a description of the YSR states in terms of a particle-in-a-box model as in [38] (now Ref. 40). To the best of our knowledge, all prior QPI measurements were done on clean surfaces or decorated with atoms only, but none on a molecular layer. The modulations in intensity in the dI/dV maps of Fig.5 may arise from the dispersion of the YSR bands. Unfortunately, it is not possible to extract meaningful information concerning the band structure. We expanded on the difficulties of k -resolved measurements in the main text.

Reviewer #3 (Remarks to the Author):

Laetitia Farinacci et al. reported on a self-assembled kagome lattice of FeP-Cl molecules on SC Pb(111) substrate, where the temperature-dependent de-chlorination was observed and used as a controlling factor, and YSR states were observed and studied by dI/dV maps. The authors tracked the hybridization of the YSR states by tuning the kagome domain sizes, and finally studied the Kondo resonance by temperature- and magnetic-field-dependent measurements after quenching the substrate's superconductivity in fields. Unfortunately, the authors claimed that flat bands as a typical feature in

kagome lattice were not possible to be experimentally observed due to the limitations of the domain size and the irregular domain boundary conditions.

Overall, this manuscript is well-written and scientifically sound. However, I would like to raise the following minor questions concerning this manuscript:

We thank the referee for their report and positive assessment of our work. We respond the individual points below. Changes to the manuscript and supplementary material are marked in red in separate files.

1. I suggest that the authors add the pristine molecular structural formula of FeP-Cl in Figure 1 as inset or somewhere else in the figures just to make it clearer and more readable.

We have followed the reviewer's recommendation and included such a molecular model in Figure 1.

2. What is the structure (or self-assemble structure) upon deposition when the substrate is held at even lower temperatures (< 230 K), where de-chlorination is not triggered? Does the pristine FeP-Cl self-assemble on Pb? If so, how is its electronic properties different compared to de-chlorinated FeP and Cl-atom kagome lattice?

In none of our preparations, we have observed the presence of islands consisting of only FeP-Cl molecules. The dechlorination process upon adsorption seems to be rather efficient and only occasionally leaves FeP-Cl molecules on the surface, such as the one of Fig2a. Presumably, Cl desorbs from the Fe center of the molecules at very low temperatures, which we cannot reach in our preparation chamber.

3. The kagome lattice pores seem to different topographic contrast, e.g., in the yellow unit cell area in Fig. 2c, three pores are grey while one pore pitch black, does this indicate different electronic states in the pores and why?

The black pitch is due to a subsurface neon impurity. These impurities are present on the clean surface [Ruby2015] and caused by Ne⁺ sputtering of the sample. They do not have a noticeable impact on the dI/dV spectra of the molecules in their vicinity. A similar Ne impurity can be seen in Fig. 2a in the lower left section. We have added a comment to the figure caption.

[Ruby2015] Ruby et al., PRL 114, 157001 (2015)

4. In Fig. 3b, the dI/dV on the central molecule (red spectrum) seems to have a huge dip at around 2 mV which is below zero, is this the sign for a negative differential resistance? If so, why is there NDR around ± 2 mV in the red spectrum?

Yes, the dip is caused by an NDR. The NDR occurs because the tip is superconducting, having sharp coherence peaks. We observe a large current when one coherence peak is aligned with another sharp resonance. This is the case for a YSR state when a resonant tunneling channel is open. A further increase of the bias voltage misaligns the YSR state and coherence peak of the tip, resulting in a decrease of the tunneling current and therefore NDR in dI/dV . The effect is the stronger the sharper is the resonance in the DoS of the sample. In order to remove this effect from the analysis of the YSR state energy, we first deconvolve the spectra and subsequently fit the sample DoS with Lorentzian

peaks. This is shown in detail in the supplementary section 3, where we show that the NDR is removed after deconvolution with the tip DoS (Fig. S3c).

5. Is there any DFT calculation results for the exchange coupling energies in this kagome lattice? How much meV are the RKKY and YSR coupling terms? What about the molecular geometrical configurations and electronic charge mapping? Is there any chemical bonding forming after annealing the sample? If so, how does the bonding affect the exchange coupling strength?

Unfortunately, there are no DFT calculations for the molecular lattice of FeP and Cl adatoms on Pb(111). While DFT calculations may be able to provide useful information on the molecular configurations and electronic charge distributions, they do not give direct information on the YSR states. This would require calculations including superconductivity, which is beyond standard DFT calculations. Such calculations are outside of the scope of our paper.

However, we note that the YSR coupling term can be estimated experimentally from the YSR splitting in Fig.4: $t \approx -0.16\text{meV}$. The strength is given by the overlap of the YSR wave functions, which can be much more extended than the spin-carrying orbitals which they originate from.

The formation of the kagome lattice requires a temperature balance which enables diffusion and Cl detachment from the Fe center while remaining below the desorption temperature of Cl. All molecular structures formed in the accessible temperature range are shown in Fig. 2. None of these structures exhibit signatures of covalent bond formation. Yet, the Cl atoms do influence the YSR states of the molecules. As described in the manuscript they effectively gate the YSR states. We speculate that this is due to an electrostatic potential, similar to the gating of YSR states in Ref. 36, where the molecular environment led to a shift of molecular orbitals and the respective YSR states.

In summary, I found the results in this manuscript solid and well-structured, but at the same time not very surprising. The key highlight is the size control of the kagome domains through ratio control of Cl/FeP by temperature, the main findings of the manuscript, i.e., the evolution of YSR states inside the SC Pb gap, are well-expected. The fact that the flat bands cannot be experimentally observed is quite a pity, which could be rather interesting using this size-variable FeP-Cl kagome lattice as a template. Considering the broad interest of readership and high prestige of Nature Communications, I personally do not feel this manuscript is fit for publication in Nature Communications. The ACS Nano could be a better venue for this manuscript. Nonetheless, I will be happy to read the revised manuscript after the authors resolve the above-mentioned issues.

REVIEWER COMMENTS

Reviewer #1 (Remarks to the Author):

The authors have addressed my comments and questions, and have updated the manuscript accordingly. As such, I recommend the publication of the manuscript in Nature Communications.

Reviewer #2 (Remarks to the Author):

The authors have addressed my comments, but I am not convinced by their answers to some of the key points. In addition, the demonstration of a band forming still remains a weak point in the manuscript as was also remarked by other referees. In summary, I do not recommend this manuscript for publication in Nature Communications.

Concerning 1), I understand the argument by the authors, but if their reasoning were correct, the system in Ref.[5] would be in the repulsive regime already at a conductance of about 5% of G_0 (about $4\mu S$ in Fig. 3ab of Ref. [5]), which is far from point contact as claimed by the authors. The system in Ref. [1] is still in the decreasing attractive regime (it has not even reached the repulsive regime) at a conductance of 30% of G_0 . The different molecules used in Ref. [1] and [5] cannot account for this drastically different behavior. Instead, it is more reasonable to assume that both systems are in the attractive regime, with a decrease in coupling for Ref. [1] and an increase in coupling for Ref. [5] as the tip approaches. The authors should take this into account in their analysis.

Concerning 2), the authors have included an analysis of the Kondo peak in a magnetic field that quenches superconductivity. Their claim is that the system is in the screened spin regime because $k_B T_K > 0.3\Delta$ according to Ref. [37]. Ref. [37] is a theoretical treatment, while Ref. [30] is an experimental measurement, which determined the quantum phase transition to be around 1.5Δ . In this case, the system would be in the free spin regime. I am wondering, which value should be the correct one to be used here, theory or experiment. Can the authors resolve this point?

Concerning 3), if I understand the authors correctly, there should be a rather symmetric YSR state half way between the red and the black dot in Fig. 3a, since there is a strong build up of intensity in

Fig. 3d. Do the authors have any spectra at these points? And along those lines, there also seems to be intensity on the chlorine atoms, which should be located in the center of the triangles. The intensity seems higher than on the black dot.

Concerning 4) through 8), I understand the authors' reasoning.

Concerning 9), I was not really talking about quasiparticle interference measurements. These would indeed be close to impossible on a molecular lattice. In analogy to the measurements and calculations done by now Ref. 40, the authors could measure the intensity of a YSR state on the center of each molecule in the 2d lattice. This could be compared with a simple tight-binding calculation mimicking the same 2d Lattice with some coupling between the states. The modulation of the intensity would give a hint at the momentum for each state. Deviations due to cotunneling other phenomena as indicated by the authors could be accounted for at least phenomenologically. The advantage here is that it is not a continuous band, but quantized states that partially follow a continuous band dispersion like in Ref. [40].

Reviewer #3 (Remarks to the Author):

The authors have satisfactorily addressed all critical issues/questions, resulting in a substantially improved manuscript. The findings in this work are scientifically sound and logically organized. However, with all due respect, I personally am still not convinced that this work has met the standards of publishing in Nature Communications regarding novelty.

Reviewer #1 (Remarks to the Author):

The authors have addressed my comments and questions, and have updated the manuscript accordingly. As such, I recommend the publication of the manuscript in Nature Communications.

We thank the reviewer for their positive assessment of our revisions and appreciate their recommendation for publication in Nature Communications.

Reviewer #2 (Remarks to the Author):

The authors have addressed my comments, but I am not convinced by their answers to some of the key points. In addition, the demonstration of a band forming still remains a weak point in the manuscript as was also remarked by other referees. In summary, I do not recommend this manuscript for publication in Nature Communications.

We appreciate that the reviewer is satisfied by most of our responses. Yet, the reviewer claims that band formation is not demonstrated in the manuscript. This is in stark contrast to the assessments of the other two reviewers, who are convinced by our arguments.

Below, we address the reviewer's remaining concerns point by point.

Concerning 1), I understand the argument by the authors, but if their reasoning were correct, the system in Ref.[5] would be in the repulsive regime already at a conductance of about 5% of G_0 (about $4\mu S$ in Fig. 3ab of Ref. [5]), which is far from point contact as claimed by the authors. The system in Ref. [1] is still in the decreasing attractive regime (it has not even reached the repulsive regime) at a conductance of 30% of G_0 . The different molecules used in Ref. [1] and [5] cannot account for this drastically different behavior. Instead, it is more reasonable to assume that both systems are in the attractive regime, with a decrease in coupling for Ref. [1] and an increase in coupling for Ref. [5] as the tip approaches. The authors should take this into account in their analysis.

We are surprised by the reviewer insisting on a comparison of two previously published papers. Furthermore, we would like to point out that the two molecules discussed in those two studies differ drastically in their structure, one of them carrying an additional oxygen ligand. The authors of Ref. [5] (reference number of SM) argue that this O atom is crucial for mediating the exchange coupling to the surface. In that regard that system is more complex. Yet, the coupling strength is also manipulated by forces exerted by the

STM tip. However, we believe that the detailed process in that publication is irrelevant to our experiments.

However, we are pleased to see that the reviewer reaches the same conclusion concerning the system of interest in this paper (namely FeP on Pb(111), previously studied in Ref [1]): at first, the tip approach takes place in the attractive regime and leads to a decrease of the coupling strength. From there follows directly that the shift towards Fermi energy indicates that the system is in the strong coupling regime before crossing of the QPT. This is exactly what has been taken into account in our analysis.

Comparison to other molecular systems is beyond the scope of our paper. Furthermore, we would like to point out that the nature of the ground state of the system is not decisive for our main observation: hybridization of YSR states.

Concerning 2), the authors have included an analysis of the Kondo peak in a magnetic field that quenches superconductivity. Their claim is that the system is in the screened spin regime because $k_B T_K > 0.3 \Delta$ according to Ref. [37]. Ref. [37] is a theoretical treatment, while Ref. [30] is an experimental measurement, which determined the quantum phase transition to be around 1.5Δ . In this case, the system would be in the free spin regime. I am wondering, which value should be the correct one to be used here, theory or experiment. Can the authors resolve this point?

As mentioned in Ref.[30], in the case of MnPc on Pb(111), the deviation of the point of the quantum phase transition from the theoretical expectation is likely due to the complex nature of the spin system, which is not a simple spin 1/2.

Here, we identify the nature of the ground state primarily by analyzing the shift of the YSR state upon tip approach, as already established in a previously published paper [31] and shown again in the SM for the molecule embedded in the kagome lattice. We then note that the point of the QPT is in agreement with the theoretical expectation (for a spin 1/2 system). The origin of different QPTs for different molecules is an interesting topic for further research, most probably involving the specific spin configuration and number of screening channels. It is beyond the scope of our paper to analyze the QPT and irrelevant for the band formation.

Concerning 3), if I understand the authors correctly, there should be a rather symmetric YSR state half way between the red and the black dot in Fig. 3a, since there is a strong build up of intensity in Fig. 3d. Do the authors have any spectra at these points? And along those lines, there also seems to be intensity on the chlorine atoms, which should be located in the center of the triangles. The intensity seems higher than on the black dot.

The evolution of the YSR state along the line between the black and red dot is shown in the Figure below. The asymmetry varies along this line and is reversed between the Fe center (brown spectrum) and ligand (blue spectrum) of the central molecule (indicated by a red dot in Fig.2 of the main text), it is symmetric between its Fe center and ligand as shown by the green spectrum as expected by the reviewer. As explained in [36], the asymmetry results from interference effects between tunneling through non-magnetic orbitals of the FeP molecule and the YSR state wave-function. At the location of the Cl atoms, the spectra show resonances as seen in the black spectrum. However, the Cl atoms themselves do not induce the YSR state. Instead as we conclude from the shape, it is the central molecule that is at the origin of the observed YSR state.

Figure: Left - topography image showing the position of the spectra displayed on the right. Spectra on the right are offset for clarity (feedback open with 200pA, 5mV)

All in all, as shown by the dI/dV maps, the YSR state is induced by the central molecule indicated by the red dot in Fig.2 of the main text and can be partially detected above the neighboring adsorbates due to the spatial extent of the YSR state.

Concerning 4) through 8), I understand the authors' reasoning.

Concerning 9), I was not really talking about quasiparticle interference measurements. These would indeed be close to impossible on a molecular lattice. In analogy to the measurements and calculations done by now Ref. 40, the authors could measure the intensity of a YSR state on the center of each molecule in the 2d lattice. This could be

compared with a simple tight-binding calculation mimicking the same 2d Lattice with some coupling between the states. The modulation of the intensity would give a hint at the momentum for each state. Deviations due to cotunneling other phenomena as indicated by the authors could be accounted for at least phenomenologically. The advantage here is that it is not a continuous band, but quantized states that partially follow a continuous band dispersion like in Ref. [40].

We agree with the referee that our experimental data can be compared with a simple tight-binding model. This is exactly what we do in Fig 4: we find a very good agreement and can unequivocally conclude that YSR states induced by neighboring FeP molecules hybridize in kagome precursors. For larger structures, while the formation of bands follows naturally from the hybridization observed in small structures several factors impede an experimental determination of the YSR dispersion (see lines 379 - 399).

In ref [40], the authors used an ingenious idea to obtain the YSR band dispersion without implementing QPI. They construct 1D chains of various lengths and match their results to the analytical solution of bound states in a 1D box. Here, our system is a 2D lattice with irregular domain boundaries that prevent a separation of the 2D Hamiltonian into a 1D problem with analytical solutions (as would be the case for, for instance, a 2D rectangular well). To obtain experimental information regarding the dispersion of the YSR bands we would need to perform QPI measurements, which are – as pointed out by the referee – very challenging.

Reviewer #3 (Remarks to the Author):

The authors have satisfactorily addressed all critical issues/questions, resulting in a substantially improved manuscript. The findings in this work are scientifically sound and logically organized. However, with all due respect, I personally am still not convinced that this work has met the standards of publishing in Nature Communications regarding novelty.

We appreciate that the referee finds our manuscript scientifically sound. However, we disagree with the statement of sufficient novelty for publication in Nature Communications. We believe that growing a kagome lattice on a superconductor while showing band formation within the superconducting energy gap is novel and sets the ground for investigating further (frustrated) lattices with competing interactions. We emphasize that (i) the superconductor prevents scattering with bulk electrons, thereby protecting the band structure from strong hybridization with bulk bands, and (ii) superconducting pairing and long-range interactions may influence the band dispersion.

REVIEWER COMMENTS

Reviewer #2 (Remarks to the Author):

The authors have replied to my concerns, but I am not quite satisfied with their responses.

Concerning 1), I have to apologize to the authors here. I actually overlooked that they use the same molecule in Ref. 1 as in this work. Maybe this could be written more explicitly in the supplementary material. Still, I find the authors' line of reasoning very peculiar here. In the previous rebuttal, they state that "Both papers (Ref. 1 and 5) cited by the referee are correct and entirely consistent with each other." In this rebuttal, the authors state that the two molecules are "drastically different" and "the detailed process [...] is irrelevant to our experiments". This does not sound very consistent and it is one reason why I was insisting on clarifying this point. The surprising point here is that the nature of the ground state is not decisive for their main observation, but the authors spend a significant part of the manuscript and the supplementary material discussing it. The authors should clarify this point in the manuscript and in the supplementary material also concerning the context of the cited references since it (obviously) leads to some confusion.

Concerning 2), I can agree with the authors that the QPT is not as universal as one might think. Now the authors reply "We then note that the point of the QPT is in agreement with the theoretical expectation (for a spin 1/2 system)." I could not find anything in the manuscript, the supplementary information or Ref. 31 that supports this statement. Where do the authors conclude this from?

Concerning 3), thank you for providing the requested data. It looks like the YSR state seems to be everywhere quite strong, but the authors mostly discuss it at the center of the molecule where it is weakest. The strongest signal is found at the ligands or in between two molecules (this is difficult to distinguish). The origin of the YSR state would typically be where the signal is strongest. Can the authors comment on that? Did the authors actually ever observe a YSR state on an isolated molecule? As the molecule is the building block and the Cl atom plays no part in the YSR state according to the authors, it would be logical to show the spectrum on a single isolated molecule. This would clarify the role of the ligands and the role of the Cl atom. Also, there is a clear YSR signal (black spectrum) on the Cl atom, but the authors write in the manuscript "The Cl atoms do not exhibit YSR resonances." This sounds contradictory and should be clarified. The molecules inside the hexagonal holes do not seem to show any YSR states. Are these the same molecules? And what about the molecules at the upper and lower triangular points in Fig. 4? They also do not seem to show any YSR states? Could it be that the Cl atoms do play a role after all, since the YSR states only seem to show up within the kagome structure, where the Cl atoms are present?

Concerning 9), do I understand the authors correctly that it is enough to show hybridization in kagome precursors to conclude that there will be band formation?

Reviewer #2 (Remarks to the Author):

The authors have replied to my concerns, but I am not quite satisfied with their responses.

Concerning 1), I have to apologize to the authors here. I actually overlooked that they use the same molecule in Ref. 1 as in this work. Maybe this could be written more explicitly in the supplementary material. Still, I find the authors' line of reasoning very peculiar here. In the previous rebuttal, they state that "Both papers (Ref. 1 and 5) cited by the referee are correct and entirely consistent with each other." In this rebuttal, the authors state that the two molecules are "drastically different" and "the detailed process [...] is irrelevant to our experiments". This does not sound very consistent and it is one reason why I was insisting on clarifying this point. The surprising point here is that the nature of the ground state is not decisive for their main observation, but the authors spend a significant part of the manuscript and the supplementary material discussing it. The authors should clarify this point in the manuscript and in the supplementary material also concerning the context of the cited references since it (obviously) leads to some confusion.

In Ref. 1 (numbering as in the Supplement) we investigated the same molecule and found a quantum phase transition by approaching it with the STM tip. In the present paper, we use the same molecule and investigate its self-assembly, in particular the formation of a kagome lattice, and we follow the hybridization from individual building blocks to extended structures with bands.

For completeness, we also discuss the spin state and its screening in the kagome assembly. To find out whether the spin is screened in the assembly or not, we make use of the same procedure of tip approach as previously published by us and in Ref. 5 (yet either, for Ref 1, on a different molecule and, for Ref 5, on the same molecular species but a different environment, namely at the edge of a kagome island). We thus cite these papers for the procedure to track the exchange coupling with the substrate. It allows to determine the ground state of the system. We believe that it is not necessary to discuss all intricacies of the molecule used in Ref. 5 to understand that our procedure is correct. We could cite only our own paper (Ref. 1), which would avoid the confusion of the referee. However, we believe that this would not be fair to the authors of Ref. 5, as they use a similar procedure and published it at almost the same time. Again, note that the molecule in Ref.5 has an additional ligand, whereas ours does not. Hence, not surprisingly, the chemical details of the bonding and, hence, the details of the interaction with the tip are different, while the overall logic is the same.

Instead of deleting the reference to Ref.5, we have clarified this point in the supplementary material section 4 and put more emphasis on the fact that our molecule is the same species as in Ref. 1.

Before: *“In order to identify the ground state of the system we perform a tip approach over the Fe center of a FeP molecule [1, 5].”*

Now: *“In order to identify the ground state of the system, one can perform a tip approach toward the molecule to modulate the exchange coupling strength of the magnetic impurity to the substrate [1,5]. The molecule studied here is the same as in [1], yet in a different molecular environment. Starting with a set point of 5mV, 200pA the approach first takes place in the attractive regime.”*

Concerning 2), I can agree with the authors that the QPT is not as universal as one might think. Now the authors reply "We then note that the point of the QPT is in agreement with the theoretical expectation (for a spin 1/2 system)." I could not find anything in the manuscript, the supplementary information or Ref. 31 that supports this statement. Where do the authors conclude this from?

This is discussed in the manuscript lines 207-213:

“This is in accordance with measurements taken in an external magnetic field, which quenches superconductivity in the Pb substrate and reveals the presence of a Kondo resonance with a Kondo temperature of $T_K = 7.9 \pm 0.3$ K (Fig. 3d). This is beyond the quantum phase transition to the screened spin regime that is expected to take place at $k_B T_K \sim 0.3\Delta$ [37].”

To be explicit, we change the last sentence (line 211-215): *“This Kondo temperature indicates a coupling strength that is beyond the quantum phase transition to the screened spin regime that is expected to take place at $k_B T_K \sim 0.3\Delta$ for a spin-1/2 impurity [37] with $\Delta=1.36$ meV.”*

Concerning 3), thank you for providing the requested data. It looks like the YSR state seems to be everywhere quite strong, but the authors mostly discuss it at the center of the molecule where it is weakest. The strongest signal is found at the ligands or in between two molecules (this is difficult to distinguish). The origin of the YSR state would typically be where the signal is strongest. Can the authors comment on that?

The origin of the YSR state is the central molecule as shown in the dI/dV maps of Fig 3. The shape and distribution of the signal in dI/dV measurement result from interfering tunneling paths, as discussed in lines 229-232:

“The shape and asymmetry of the YSR state in dI/dV measurements are related to interfering tunneling paths through the magnetic and frontier orbitals of the FeP molecule [36]. Essentially, the shape seen in the dI/dV maps thus resembles the molecular orbitals rather than the wave function of the YSR states in the substrate.”

Did the authors actually ever observe a YSR state on an isolated molecule? As the molecule is the building block and the Cl atom plays no part in the YSR state according to the authors, it would

be logical to show the spectrum on a single isolated molecule. This would clarify the role of the ligands and the role of the Cl atom.

Indeed, we observe molecules that are unaffected by Cl.

Here we show an isolated molecule on a clean terrace. It exhibits asymmetric BCS coherence peaks with the asymmetry inverting from the ligand to the center (see Fig. 1, below). We attribute the asymmetry to YSR states close to the gap edge. However, these molecules are not stable on the surface as they rotate under the influence of the STM tip. Hence, it is not meaningful to map out the distribution of the YSR state. However, the observed asymmetry is in agreement with the observation on molecules found in the pores of the kagome lattice that we described in the paper.

Figure 1. Isolated molecules show a YSR state close to the gap edge. The black spectrum is taken on Pb for reference. The red and yellow spectra are offset for clarity. Set point: 5mV, 200pA

More specifically, spectra on molecules that reside inside the pores of the kagome lattice are described in connection with Fig. 5 (blue spectrum), towards the end of the manuscript, lines 338-344:

“In agreement with the observations above, molecules that are not surrounded by two Cl atoms exhibit sharp and energetically isolated states. For example, a molecule inside a pore of the kagome lattice (blue spectrum) shows a YSR state close to the coherence peaks (compare to isolated molecules in Fig. 3).”

We now also add a reference to the later description when discussing the FeP molecules, lines 220-226:

“Molecules that are isolated on the surface (see SM, section 7) or lie within a pore of the kagome lattice as displayed below in Fig. 5d) also show only a weak asymmetry of the BCS coherence peaks. These types of spectra indicate that a single Cl atom does not strongly influence the position of the YSR states, in contrast to the molecules with two Cl neighbors.”

We also explicitly add a sentence at the end of the paragraph instead of one paragraph later, lines 235-239: "The Cl atoms do not induce YSR states. Instead they act as local gates modifying the coupling strength similar to electrostatic fields imposed by neighboring molecules [32] or modulations in electronic potential due to a charge-density wave [38]."

To further highlight the role of Cl, we also presented data in Supplementary Fig.S2 when only one Cl atom is next to a FeP molecule.

Also, there is a clear YSR signal (black spectrum) on the Cl atom, but the authors write in the manuscript "The Cl atoms do not exhibit YSR resonances." This sounds contradictory and should be clarified.

We meant to say that the Cl adatoms do not induce YSR resonances, as the resonances in the black spectrum originate from the neighboring FeP molecule. We have amended the sentence:

Before: "*The Cl atoms do not exhibit YSR resonances.*"

Now: "*The Cl atoms do not induce YSR states.*"

The molecules inside the hexagonal holes do not seem to show any YSR states. Are these the same molecules?

The molecules inside the hexagonal holes show a YSR state close to the gap edge, as shown in the blue spectrum of Fig.5d and discussed lines 341-343:

"a molecule inside a pore of the kagome lattice (blue spectrum) shows a YSR state close to the coherence peaks"

We have elaborated on this fact in more detail above.

And what about the molecules at the upper and lower triangular points in Fig. 4? They also do not seem to show any YSR states?

Molecules at the edge of kagome precursors are discussed in section 2 of the supplementary material and Fig.3 of the main text. They only have one Cl adatom in their vicinity and show an asymmetry of the coherence peaks (manuscript: Fig3b, black spectrum, supplementary: Fig S2) attributed to the presence of a YSR state close to the gap edge.

Could it be that the Cl atoms do play a role after all, since the YSR states only seem to show up within the kagome structure, where the Cl atoms are present?

The Cl atoms play a role in tuning the energy of the YSR states. We have made this point more clear in the main text (see also response above) lines 235-239.

“The Cl atoms do not induce YSR states. Instead they act as local gates modifying the coupling strength similar to electrostatic fields imposed by neighboring molecules [32] or modulations in electronic potential due to a charge-density [38].”

Concerning 9), do I understand the authors correctly that it is enough to show hybridization in kagome precursors to conclude that there will be band formation?

Yes, the observation of hybridization in structures of a few units will ultimately lead to band formation in infinite structures in agreement with the broad resonances on the extended structures.

REVIEWERS' COMMENTS

Reviewer #2 (Remarks to the Author):

The authors have made a great effort to clarify my points over several rounds now and I am ready to accept the manuscript. The only remaining point that arises is the asymmetry in the coherence peaks in Fig. 1 of the rebuttal, which the authors interpret as YSR states that are very close to the gap edge. However, the signal outside of the gap (normal conducting part) seems to scale in the same way as the coherence peaks, i.e. it shows a very similar asymmetry, which calls the interpretation as YSR states into question. Only in the reference spectrum on Pb the coherence peaks are symmetric along with the normal conducting part. I will admit though that this is a close call and it will probably be difficult to distinguish these scenarios.

Reviewer #2 (Remarks to the Author):

The authors have made a great effort to clarify my points over several rounds now and I am ready to accept the manuscript. The only remaining point that arises is the asymmetry in the coherence peaks in Fig. 1 of the rebuttal, which the authors interpret as YSR states that are very close to the gap edge. However, the signal outside of the gap (normal conducting part) seems to scale in the same way as the coherence peaks, i.e. it shows a very similar asymmetry, which calls the interpretation as YSR states into question. Only in the reference spectrum on Pb the coherence peaks are symmetric along with the normal conducting part. I will admit though that this is a close call and it will probably be difficult to distinguish these scenarios.

We thank the reviewer for their positive assessment of our revisions and are pleased that they are now ready to accept the manuscript. Concerning this last point, the asymmetry of the normal conducting part of the spectra is due to the background conductance of the molecules in this bias window. More precisely, molecules with one Cl neighbour display a Kondo resonance (see molecules with blue crosses in Fig.6 of the main text, and Supplementary Notes 5 for more details), which is detected by interfering tunnelling paths (see ref. 36) leading to the observed asymmetry in dI/dV . Since we can correlate the presence of this resonance to that of a YSR state – sometimes well-separated from the coherence peaks as for molecules with two Cl neighbours, sometimes very close to the gap edge as in Supplementary Figure 8 where a small shift in energy is detected along the with the asymmetry of the coherence peaks – we attribute the asymmetry of the coherence peaks to the presence of a YSR state close to the gap edge, which we detect with limited energy resolution.